# Leave one Expert Out: Robust Uncertainty Quantification via Intrinsic Cross-Validation

## Abstract

Estimating epistemic uncertainty remains an important challenge in modern Deep Learning (DL). We propose a novel architecture, called Leave one Expert Out (LEO), which is a form of a mixture-of-experts model with latent-space-distance-aware router and a null expert, representing prior belief, to which output of the model collapses if testing datapoint is too different from any of datapoints experts were trained on. This architecture allows to temporarily drop experts from the model, and we utilise this property to train the router to leverage the predictions of remaining experts to make predictions for the datapoints normally assigned to the expert currently removed from the model. We coin this mechanism *intrinsic cross-validation* and show, such a trained router excels at estimating epistemic uncertainty for both in and out of distribution inputs. We demonstrate state-of-art performance on uncertainty quantification in regression benchmarks, such as UCI problems or age prediction on UTK-Face, and CIFAR-10 classification benchmark. We also show the proposed method can achieve superior performance in surrogate-based black-box optimization.

## 1 Introduction

Deep Learning (DL) (Rumelhart et al., 1986; Goodfellow et al., 2016) has achieved spectacular success when it comes to the predictive power of models. Beginning with early successes in computer vision (Krizhevsky et al., 2012), where the models were trained to predict the class of an object in an image, the field has since advanced rapidly. Today, modern DL models, such as Large Language Models, can even engage in meaningful conversations with the user by predicting the most likely next word (token) given a sequence of preceding words. However, while DL models excel at making a prediction, assessing the certainty of that prediction remains a notoriously difficult problem.

This uncertainty might stem from different sources. Aleatoric uncertainty reflects inherent noise in the data or labels. For example, the same house might sell for slightly different prices due to random factors not captured by its features. In general, basic DL models can typically handle this type of uncertainty if their outputs can be interpreted as probability distributions. For instance, in classification with softmax outputs, if two identical images exist in the training set, but the first is labelled as a dog and second as a cat, then training with standard cross-entropy loss will encourage the model to put roughly half of probability mass on each of the labels. While more sophisticated techniques exist for modelling aleatoric uncertainty, even simple models provide a basic way to capture this type of observation noise.

The second source of uncertainty is typically much harder to deal with. It is referred to as epistemic and arises when the model has not seen enough data during training to make a confident prediction for a given test data point. We cannot simply train the model to output its estimated epistemic uncertainty, because all training points are in-distribution (ID) and this uncertainty during training is essentially zero (or very small and only due to observation noise). As a result, naively trained models tend to be overconfident and behave unpredictably on inputs far from the training data. Since epistemic uncertainty reflects a model's lack of knowledge about new inputs, a proper model of epistemic uncertainty must, by definition, account for out-of-distribution (OoD) inputs.

At first glance, this problem may seem prohibitively difficult. How can we make sure our epistemic uncertainty model performs well on inputs it has never seen? But if we take a step back, and

consider the classical, non-deep machine learning methods, we will realize that this exact problem has already been addressed countless times. One of the classical models celebrated for uncertainty quantification is the Gaussian Process (GP) (Rasmussen & Williams, 2006). GPs have the rather desirable property that, as the inputs move further away from training data, the model predictions collapse to the user-specified prior, with the rate of collapse controlled by the length scale hyperparameter. This hyperparameter can be tuned using cross-validation (Bachoc, 2013), where the model is repeatedly trained on subsets of the data and evaluated on held-out points. While feasible for classical models with short training times, repeated retraining is completely impractical for large deep learning models, requiring plenty of time and compute to retrain.

In recent years, numerous uncertainty quantification methods have been developed for epistemic uncertainty quantification, including Bayesian neural networks (Mackay, 1992; Neal, 2012), mean-field variational inference (Blundell et al., 2015), Monte Carlo Dropout (Gal & Ghahramani, 2016), ensembles Lakshminarayanan et al. (2017); Wen et al. (2020); Dusenberry et al. (2020) and single-model approaches (Tagasovska & Lopez-Paz, 2019; Van Amersfoort et al., 2020; Liu et al., 2020; Van Amersfoort et al., 2021). However, in these approaches, the training process typically does not explicitly encourage the model to outputs high uncertainty in OoD cases. Instead, they rely on the assumption that the model will naturally behave differently on OoD inputs, which does not necessarily need to hold in practice.

In this work, we propose a novel approach for epistemic uncertainty quantification called Leave-one-Expert-Out (LEO). LEO introduces supervised OoD signals during training by simulating OoD scenarios using only partitioned training data, without requiring actual held-out OoD examples. The intuition is that this enables the model to transfer its OoD detection capability to test time. LEO is a variant of a mixture-of-experts neural network, where each expert is trained on a subset of the training data, and OoD scenarios are simulated by randomly dropping some experts during training.

Unlike some methods that set a fixed threshold to reject model outputs for OoD inputs, we treat all unseen inputs as "partially" OoD. To handle this, we include a "null" (prior) expert, outputting a vague distribution suitable for OoD cases. Predictions from this expert and the other experts are then weighted by a distance-aware router, which computes weights based on the distance between the test input and the training data in the latent space. To train this router, we introduce a novel mechanism called "intrinsic cross-validation", which involves learning to make accurate predictions for data assigned to a given expert, *with that expert removed from the model*. This forces the router to learn how much to rely on the remaining experts' predictions and when to defer to the null expert, mimicking the desirable property of a GP.

In our architecture, experts can share the feature extractor and differ only in the final layer, resulting in a negligible increase in model size. Through extensive experiments, we show that LEO obtains superb performance on both regression and classifications tasks requiring uncertainty quantification, for both in- and out-of-distribution data, as well as sequential decision making, ~~often completely outperforming existing methods~~ and consistently matches or outperforms state of art performance.

## 2 METHODOLOGY

This section presents the core mechanism of LEO. The training and inference procedures are summarised in Algorithms 1, 2 and 3. We consider a supervised learning problem where, given a point $x \in \mathcal{X}$, the goal is to predict the target $y \in \mathcal{Y}$. We assume that we are given a training dataset $\mathcal{D} = \{(x_i, y_i, )_{i=1}^n\}$, where $x_i \in \mathcal{X}$ are inputs and $y_i \in \mathcal{Y}$ are labels. The set $\mathcal{Y}$ could, for instance, be $\mathbb{R}$ for regression or $[C]$ for $C$-class classification. We aim to devise an architecture that

  (i) produces epistemic uncertainty estimates that grow as the input moves further away from the training distribution;

  (ii) enables efficient cross-validation of the uncertainty estimates.

To fulfill these requirements, we propose to use a mixture-of-experts style architecture, described below. Before training, we assign every data point in the training set a type $t \in \mathcal{T}$, with each type handled by a specific expert. We discuss how to partition the training set into different types in Subsection B.1. Let $\mathcal{E} \subset \mathcal{T}$ be the set of all data point types in the training data. We propose to use a shared feature extractor $f : \mathcal{X} \to \mathcal{Z} \subset \mathbb{R}^{d_z}$ parameterised by $\psi$ and implement each expert

for $t \in \mathcal{E}$ as a linear head[1] operating on the latent representation $z = f(x; \psi)$. As such, each expert is a single-layer network $h_t : \mathcal{Z} \to \mathbb{R}^{d_y}$, where $d_y = 1$ for regression and $d_y = C$ for $C$-class classification. Each expert is trained only on data points of its own type and the feature extractor is trained on all the data. That is, we learn $\{\theta_t\}_{t \in \mathcal{E}}, \psi$ by minimizing the following loss function:

$$\mathcal{J}^{\text{experts}}(\mathcal{D}; \{\theta_t\}_{t \in \mathcal{E}}; \psi) = \frac{1}{n} \sum_{t \in \mathcal{E}} \sum_{\{i : t_i = t\}} \mathcal{L}(h_t(f(x_i; \psi); \theta_t), y_i),$$

where $\mathcal{L}(\cdot, \cdot) : \mathbb{R}^{d_y} \times \mathcal{Y} \to \mathbb{R}$ is a task-specific loss function. We use Mean Squared Error for regression and Cross-Entropy for classification. Note that, although the feature extractor technically sees all the data points, the predictions made by each expert can vary significantly if each expert only sees data from a particular subregion of $\mathcal{X}$. We expand on this in Subsection B.1.

When given a new data point unseen during training, we do not know a priori which expert will handle it best. Hence, during inference, which expert to invoke is decided by a router $p_\phi(t|x; \mathcal{E})$ parameterised by $\phi$. We thus make the prediction for a new point by marginalizing the type variable:

$$p_\phi(y|x; \mathcal{E}) = p_0(y)p_\phi(t \notin \mathcal{E}|x; \mathcal{E}) + \sum_{t \in \mathcal{E}} p(y|x, t)p_\phi(t|x; \mathcal{E}), \tag{1}$$

where $p_0$ is a prior distribution associated with an additional out-of-distribution (OoD) type. The notation $t \notin \mathcal{E}$ is thus shorthand for this OoD type, i.e., the case where none of the experts associated with the types $\mathcal{E}$ is expected to provide an accurate prediction. The resulting prediction can be interpreted as a weighted mixture of the experts' in-distribution predictions and the prior distribution $p_0$, where the weight assigned to $p_0$ reflects the model's estimated probability of the input being OoD. This prior can be specified by the user if they have domain knowledge about the distribution of $y$. In our experiments, we simply resort to a uniform distribution over all classes in the case of classification and a zero-mean, unit-variance gaussian in the case of regression (and we assume that the training data is standardized). For the predictive distribution $p(y|x, t)$ in Equation 1, we use the predictive softmax $p(y|x, t) = \text{softmax}(h_t(z; \theta_t))$ in classification and the delta function centered on the expert's prediction $p(y|x, t) = \delta(y = h_t(z; \theta_t))$ in regression. As such, the uncertainty in this model mainly arises when $p_\phi(t \notin \mathcal{E}|x; \mathcal{E})$ is high, in which case the vague prior dominates.

In the case of classification, the final predictive distribution $p_\phi(y|x; \mathcal{E})$ is just a mixture of categorical distributions, which is a categorical distribution itself that can be easily computed. In regression, given the prior $p_0(y|x) = \mathcal{N}(y; \mu_0(x), \sigma_0^2(x))$, then $p_\phi(y|x; \mathcal{E})$ is a mixture of a Gaussian and delta functions, which we approximate with a single Gaussian by moment-matching, i.e., $p_\phi(y|x; \mathcal{E}) \approx \mathcal{N}(y; \mu_\phi(x; \mathcal{E}), \sigma_\phi^2(x; \mathcal{E}))$, where

$$\mu_\phi(x; \mathcal{E}) = \mu_0(x)p_\phi(t \notin \mathcal{E}|x; \mathcal{E}) + \sum_{t \in \mathcal{E}} h_t(f(x; \psi); \theta_t) \, p_\phi(t|x; \mathcal{E}),$$

$$\sigma_\phi^2(x; \mathcal{E}) = \left(\sigma_0^2(x) + (\mu_0(x) - \mu_\phi(x; \mathcal{E}))^2\right) p_\phi(t \notin \mathcal{E}|x; \mathcal{E}) + \sum_{t \in \mathcal{E}} \left(h_t(f(x; \psi); \theta_t) - \mu_\phi(x; \mathcal{E})\right)^2 p_\phi(t|x; \mathcal{E}).$$

We now proceed to describe the mechanism behind the operation of the router.

## 2.1 DISTANCE-AWARE ROUTER

The router, which models the type probabilities $p_\phi(t|x; \mathcal{E})$ and $p_\phi(t \notin \mathcal{E}|x; \mathcal{E})$, operates on the latent embeddings given by the feature extractor $f(x; \psi)$ and is parametrised by $\phi = \left(\bigcup_{t \in \mathcal{E}} \phi_t\right) \cup \phi_0$, where $\phi_t$ is a set of parameters specified below for each $t$, and $\phi_0 \in \mathbb{R}$ is a learnable constant. To fulfill the requirement (i) outlined in the beginning of the section, we want the router to be distance-aware in the latent space, i.e., to guarantee that a data point with latent embeddings $z$ vastly different than ones seen during training will make the router output a high OoD probability $p_\phi(t \notin \mathcal{E}|x; \mathcal{E})$ and make the predictive distribution collapse to the prior $p_0$. To achieve this property, we propose that the router should learn a projection matrix $M_t$ for each expert for $t \in \mathcal{E}$ and assign a score inversely proportional to the L2 distance between the projected embeddings $z^T M_t$ and the centroid

---

[1]In principle, each expert head could be much deeper than a single layer. However, we found empirically that a single layer was sufficient, so we chose it for simplicity and to reduce memory and computational costs.

$e_t$ of the data points of type $t$ in the latent space:

$$s_t(z; \phi_t) = \frac{\tau_t}{\frac{1}{d_z}\|z^T M_t - e_t\|_2^2},$$

where $\tau_t$ is a temperature parameter and $\phi_t = \{M_t, \tau_t\}$. Since the function $\|z^T M_t - e_t\|_2^2$ is quadratic and always positive, it has a unique minimiser and for any direction $\hat{e} \in \mathbb{R}^{d_z}$, we must have $s_t(\alpha\hat{e}; \phi_t) \to 0$ as $\alpha \to \infty$. The use of such distances was previously introduced in a method called DUQ (Van Amersfoort et al., 2020), which uses the exponent of the negative distance rather than the inverse distance. In our experiments, we found the inverse distance to be a much more stable choice for the router. The scores are then normalised as below to give the type probabilities:

$$p_\phi(t|x; \mathcal{E}) = \begin{cases} \frac{s_t(f(x;\psi);\phi_t)}{\phi_0 + \sum_{t' \in \mathcal{E}} s_{t'}(f(x;\psi);\phi_{t'})} & \text{for } t \in \mathcal{E} \\ \frac{\phi_0}{\phi_0 + \sum_{t' \in \mathcal{E}} s_{t'}(f(x;\psi);\phi_{t'})} & \text{for } t \notin \mathcal{E}. \end{cases} \tag{2}$$

When the latent embedding $z$ for a given point becomes too distant from the embeddings seen during training, we have $s_t(\alpha\hat{e}; \phi_t) \to 0$ for all $t \in \mathcal{E}$, as explained above. In this case, the constant $\phi_0$ must necessarily start to dominate and $p_\phi(t \notin \mathcal{E}|x; \mathcal{E}) \to 1$. This fulfills the requirement (i) outlined at the beginning of this section, but requirement (ii) is still not addressed. Indeed, to make the uncertainty estimate meaningful, it is necessary to determine how fast $p_\phi(t \notin \mathcal{E}|x; \mathcal{E})$ collapses to 1, to make sure that in-distribution data for which we can still make valid predictions are assigned relatively small uncertainty and out-of-distribution data for which we cannot hope to make good predictions are given high uncertainty. In the next subsection, we expand on how to achieve this by leaving an expert out, a powerful mechanism that the proposed architecture allows us to exploit.

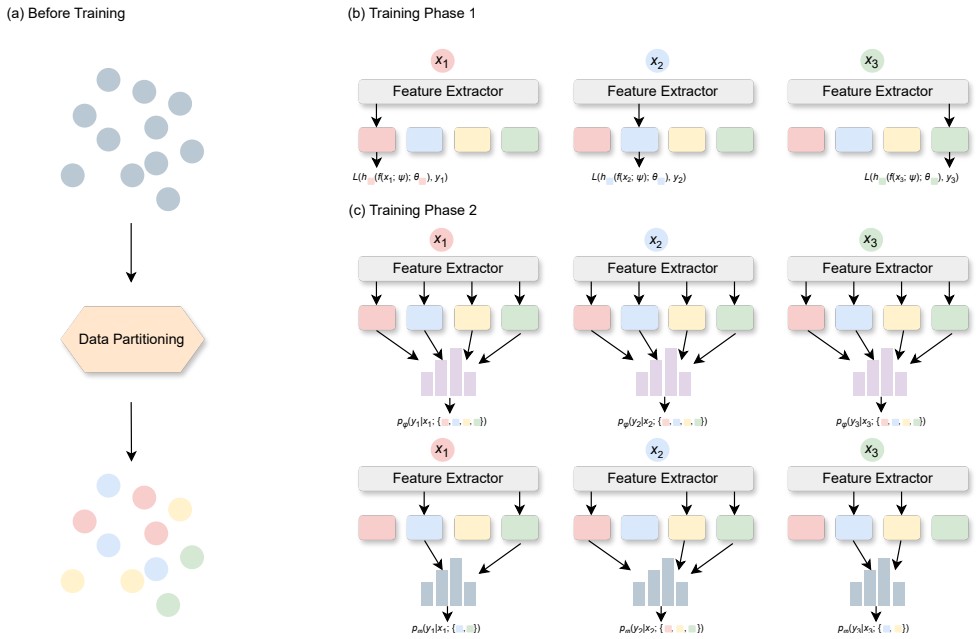

Figure 1: (a) Before training, the dataset is partitioned into different types. (b) Training Phase 1: Each data point is passed through the feature extractor and the type-specific expert to optimise per-expert MSE losses. Both the feature extractor and experts are updated. (c) Training Phase 2: The feature extractor and experts are frozen, and only the router is updated. In the top panel, each data point is passed through all experts, with outputs weighted by the router to compute the likelihood under the full model. In the bottom panel, the expert corresponding to the data point's type and a random subset of other experts are dropped, and the remaining experts' weighted outputs are used to compute the intrinsic cross-validation likelihood. Both likelihoods are obtained from a single forward pass by using different subsets of experts.

## 2.2 LEAVE-ONE-EXPERT-OUT: INTRINSIC CROSS-VALIDATION

We are now going to discuss a crucial mechanism of the Leave-one-Expert-Out (LEO) architecture. Note that in Equation 1, we used the notation $p_\phi(y|x; \mathcal{E})$, which means that the final output distribution of the full model depends on the all the known data types $\mathcal{E}$. Let us consider what will happen if we evaluate this equation with some type $t^\star$ removed from $\mathcal{E}$:

$$p_\phi(y|x; \mathcal{E} \setminus t^\star) = p_0(y)p_\phi(t \notin \mathcal{E} \setminus t^\star|x; \mathcal{E} \setminus t^\star) + \sum_{t \in \mathcal{E} \setminus t^\star} p(y|x, t)p_\phi(t|x; \mathcal{E} \setminus t^\star).$$

First of all, the term corresponding to $t^\star$ is now omitted entirely and the output $h_{t^\star}(y|x)$ of the corresponding expert does not contribute to the final model output. Secondly, the prior $p_0(y)$ is now multiplied by the probability of the data point being of type $t \notin \mathcal{E} \setminus t^\star$, as opposed to $t \notin \mathcal{E}$. As such, the model behaves as if none of the data points of type $t^\star$ had been seen during training. In other words, when the type $t^\star$ is dropped, the set of known data types becomes $\mathcal{E} \setminus t^\star$, and all other types are treated as OoD. In this case, $p_\phi(\cdot|x; \mathcal{E} \setminus t^\star)$ is defined in a similar way to Equation 2, but with the score function $s_{t^\star}(z; \phi_{t^\star})$ replaced by $\phi_0$. We train the router by minimising the loss:

$$\mathcal{J}^{\text{router}}(\mathcal{D}; \phi) = -\Big( \log p_\phi(\mathcal{D}) + \log p_\phi(\mathcal{D}_{\text{ICV}}) \Big),$$

where ICV stands for *intrinsic cross-validation*, which we define below. The parameters $\theta$ of the experts and the parameters $\psi$ of the feature extractor are kept frozen (detached), so only the parameters of the router $\phi$ are updated when $\mathcal{J}^{\text{router}}$ is optimised. We updated centroids $e_t$ in the same way as done in DUQ; see Appendix B.3 for details. In the first term, $p_\phi(\mathcal{D})$ denotes the likelihood of the data under the full model defined in Equation 1 without dropping any of the experts. For completeness, we provide the expression below:

$$p_\phi(\mathcal{D}) = \prod_{i=1}^{n} p_\phi(y_i|x_i; \mathcal{E}).$$

The role of this term in the loss function is to make sure that the predictions of all experts are meaningfully combined by the router and lead to a sensible data fit. However, this term on its own does not guarantee sensible uncertainty quantification. In fact, if each expert can model its data points perfectly, the optimal solution is just to always collapse the probability on that expert, resulting in no uncertainty quantification. This is remedied by $p_\phi(\mathcal{D}_{\text{ICV}})$ in the second term, which we call the *intrinsic cross-validation* likelihood. It involves making a prediction for each data point $i$, with the expert for type $t_i$ and some random subsets of other experts dropped from the model, i.e.,

$$p_\phi(\mathcal{D}_{\text{ICV}}) = \prod_{i=1}^{n} p_\phi(y_i|x_i; \mathcal{E} \setminus (t_i \cup r(\mathcal{E}))),$$

where $r(\mathcal{E})$ is a randomly selected subset of $\mathcal{E}$. See Figure 1. We describe how exactly this random subset is selected in Subsection B.2. Note that if an expert that is not dropped from the model can extrapolate well to data point types that it did not see during training, the router can achieve a good intrinsic cross-validation likelihood $p_\phi(\mathcal{D}_{\text{ICV}})$ by putting a high probability mass on it, e.g., by setting its temperature $\tau_t$ high. Conversely, if each of the remaining experts makes a wrong prediction, collapsing to the vague prior $p_0$ will be the optimal solution. As such, the router needs to learn its parameters to find the optimal rate at which the model stops trusting the known experts and collapses to the prior, effectively learning how to estimate its epistemic uncertainty.

## 3 RELATED WORK

**Epistemic uncertainty and out-of-distribution detection** Ensemble methods (Lakshminarayanan et al., 2017; Wen et al., 2020; Dusenberry et al., 2020; Zaidi et al., 2021) are a standard approach for estimating epistemic uncertainty, combining predictions from multiple independently trained models. Monte Carlo Dropout (Gal & Ghahramani, 2016) offers a lightweight alternative by applying dropout at test time and averaging multiple forward passes. While ensembles remain state-of-the-art, they are computationally expensive as both training and inference scale with the number of models. Moreover, theoretical work has questioned whether ensembles truly capture epistemic uncertainty or primarily reflect randomness in initialization and optimisation (He et al., 2020).

Single-model methods require only a single forward pass at test time. Distance-based approaches such as deterministic uncertainty quantification (DUQ) (Van Amersfoort et al., 2020), spectral-normalized neural Gaussian processes (SNGP) (Liu et al., 2020), and deterministic uncertainty estimation (DUE) (Van Amersfoort et al., 2021) use distance-aware output layers (e.g., RBFs, GPs) to improve OoD sensitivity together with spectrally normalised (Miyato et al., 2018) or gradient penalised (Van Amersfoort et al., 2020) feature extractor. Distributional approaches, including evidential deep learning (EDL) (Sensoy et al., 2018; Amini et al., 2020) and Density Regression (DR) (Bui & Liu, 2024), model predictive distributions directly without requiring sampling. Bayesian Neural Networks (BNNs) are a broad family of approaches for assesing uncertainty in NNs and involve methods such as Bayes-by-Backprop (Blundell et al., 2015), Laplace Approximation and Variational Inference (Wen et al., 2018). However, these approaches typically rely on extensive sampling and suffer from instabilities. Variational Bayes last layer (VBLL) (Harrison et al., 2024) is a recent, state-of-the-art BNN approach that applies Bayesian inference only to the final layer, avoids sampling all-together and enjoying much more stable performance. Although single-model approaches may not always match ensembles in performance, they provide efficient alternatives suitable for large-scale deployment. Epistemic neural networks ("epinets") (Osband et al., 2023) represent a related direction by conditioning predictions on an auxiliary epistemic index. In comparison, LEO modifies the final layer with a mixture-of-experts structure and addresses OoD detection via training-time OoD simulations.

Although this work focuses on supervised learning, OoD detection has also been studied in generative modeling. Prior work has shown that deep generative models can assign high likelihoods to OoD data (Nalisnick et al., 2018; Choi et al., 2018; Kirichenko et al., 2020), raising concerns about using density estimates from generative models for OoD detection. Alternative strategies include hypothesis testing frameworks (Nalisnick et al., 2019) and training with auxiliary OoD datasets (Hendrycks et al., 2018). In contrast, LEO does not require a separate OoD dataset and can simulate OoD situations using training set only via the mechanism of intrinsic cross-validation.

**Mixture-of-Experts models**  Mixture-of-Experts (MoE) models (Jacobs et al., 1991; Jordan & Jacobs, 1994) divide a prediction task among multiple specialized sub-networks, or experts, with a gating function that determines how to combine their outputs. Experts can share feature representations, allowing increased model capacity with minimal additional parameters. LEO builds on this framework by leveraging the experts to capture epistemic uncertainty and including a "null" expert to represent lack of confidence.

## 4 EXPERIMENTS

We evaluate our algorithm LEO together with baselines on uncertainty quantification in regression and classification tasks, as well as on Bayesian Optimisation (BO) tasks, where the goal is to sequentially query an unknown black-box function to find points with the highest objective values. In all tasks, except for BO, we reserve 10% of the training data as a validation set and apply early stopping based on the validation log-likelihood. We now describe the baselines used in our experiments. We share our code via an anonymysed link[2].

**Baselines** For comparison, we selected the strongest existing uncertainty quantification baselines. These include MC Dropout (Gal & Ghahramani, 2016), Ensemble (Lakshminarayanan et al., 2017), EDL (Sensoy et al., 2018; Amini et al., 2020), DUE (Van Amersfoort et al., 2021) and VBLL (Harrison et al., 2024). Additionally, in all regression and BO tasks we compare against Density Regression (Bui & Liu, 2024) and in all classification tasks we compare against DUQ (Van Amersfoort et al., 2020). We try to make the setup and architectures as similar across baselines as possible; see Appendix C for details.

---

[2]https://anonymous.4open.science/r/leave-one-expert-out-DF01/

Table 1: Results for four UCI benchmarks. Reported values are means over 20 seeds and the values after ± denote 95% CIs of the mean estimator. The best methods and all methods that do not statistically differ w.r.t. two-sided z-test are shown in bold. The second best methods are underlined.

| Dataset | kin8nm | | naval | | power-plant | | yacht | |
|---|---|---|---|---|---|---|---|---|
| Metric | NLL (↓) | R2 (↑) | NLL (↓) | R2 (↑) | NLL (↓) | R2 (↑) | NLL (↓) | R2 (↑) |
| Density R. | 0.18 ± 0.03 | 0.92 ± 0.00 | -2.24 ± 0.05 | 1.00 ± 0.00 | **-0.09 ± 0.02** | 0.95 ± 0.00 | 1.27 ± 1.20 | 0.99 ± 0.00 |
| Dropout | 1.19 ± 0.12 | 0.92 ± 0.00 | -1.12 ± 0.02 | 0.99 ± 0.00 | 3.13 ± 0.34 | **0.96 ± 0.00** | -1.23 ± 0.27 | 0.98 ± 0.00 |
| DUE | 1.95 ± 0.12 | 0.80 ± 0.01 | -0.36 ± 0.30 | 1.00 ± 0.00 | 1.20 ± 0.09 | 0.89 ± 0.00 | -1.49 ± 0.05 | **1.00 ± 0.00** |
| EDL | 0.18 ± 0.03 | 0.91 ± 0.01 | -1.84 ± 0.03 | 1.00 ± 0.00 | **-0.09 ± 0.04** | 0.95 ± 0.00 | -2.07 ± 0.34 | 0.99 ± 0.00 |
| Ensemble | 1.32 ± 0.19 | **0.93 ± 0.00** | -2.26 ± 0.04 | **1.00 ± 0.00** | 1.72 ± 0.25 | **0.96 ± 0.00** | -2.51 ± 0.33 | **1.00 ± 0.00** |
| VBLL | 2.75 ± 1.99 | 0.89 ± 0.00 | -0.53 ± 0.25 | 0.99 ± 0.00 | -0.04 ± 0.04 | 0.95 ± 0.00 | 0.03 ± 0.93 | 0.99 ± 0.00 |
| LEO (ours) | **0.12 ± 0.01** | 0.92 ± 0.00 | **-2.62 ± 0.08** | **1.00 ± 0.00** | **-0.04 ± 0.05** | 0.95 ± 0.00 | -2.16 ± 0.23 | 0.99 ± 0.00 |

Table 2: Results for UCI protein and UTK-Face benchmarks. Reported values are means over 20 seeds in protein and 5 seeds in UTK, with the values following ± denoting 95% CIs of the mean estimator. The best-performing methods and those tied via a z-test are shown in bold, while the second-best methods are underlined. In cells marked with (*), predictive variance was so small that likelihood computations caused a numerical issue on all seeds.

| Dataset | protein | | | | UTK | | | |
|---|---|---|---|---|---|---|---|---|
| Metric | NLL (↓) | R2 (↑) | OOD NLL (↓) | OOD R2 (↑) | NLL (↓) | R2 (↑) | OOD NLL (↓) | OOD R2 (↑) |
| Density R. | **1.01 ± 0.24** | 0.59 ± 0.01 | 11.52 ± 2.50 | 0.39 ± 0.05 | 1.23 ± 0.11 | 0.65 ± 0.01 | **0.49 ± 0.07** | 0.53 ± 0.04 |
| Dropout | 4.17 ± 0.35 | **0.69 ± 0.00** | 4.86 ± 0.41 | **0.53 ± 0.01** | N/A(*) | 0.75 ± 0.02 | N/A(*) | **0.55 ± 0.09** |
| DUE | 5.11 ± 0.21 | 0.08 ± 0.01 | 2.99 ± 0.12 | 0.14 ± 0.01 | 1.73 ± 0.22 | 0.00 ± 0.00 | 1.24 ± 0.08 | -0.70 ± 0.01 |
| EDL | 1.07 ± 0.02 | 0.41 ± 0.01 | **1.15 ± 0.08** | 0.44 ± 0.03 | 2.36 ± 0.24 | 0.62 ± 0.06 | 2.27 ± 0.27 | 0.14 ± 0.38 |
| Ensemble | 2.27 ± 0.11 | 0.68 ± 0.00 | 1.90 ± 0.21 | 0.28 ± 0.05 | 1.10 ± 0.13 | 0.79 ± 0.01 | 0.68 ± 0.16 | **0.60 ± 0.05** |
| VBLL | 1.00 ± 0.03 | 0.59 ± 0.01 | 2.31 ± 0.36 | -0.18 ± 0.14 | **0.85 ± 0.27** | 0.82 ± 0.01 | 0.79 ± 0.32 | 0.59 ± 0.07 |
| LEO (ours) | **0.89 ± 0.04** | 0.60 ± 0.01 | **1.19 ± 0.05** | 0.42 ± 0.02 | **0.78 ± 0.03** | 0.74 ± 0.02 | 0.49 ± 0.09 | 0.58 ± 0.06 |

## 4.1 Regression Problems

To evaluate performance on regression tasks, we consider ten UCI benchmarks and the UTK-Face dataset, where the goal is to predict age from raw pixels of facial images. For each dataset, we report the negative log-likelihood (NLL, lower is better), and coeff. of determination ($R^2$, higher is better) or mean absolute error (MAE, lower is better) as a measure of predictive performance, depending on the task. We detail how OOD evaluation sets were obtained in Appendix D.

**Results** We present results on five UCI datasets and UTK-Face in Tables **??** and 2, and defer the rest of UCI datasets to Appendix F due to space limitations. Overall, we can see that LEO excels in terms of NLL, achieving the best (or tying for the best) performance across all of the evaluated regression benchmarks. Among the remaining baselines, methods such as EDL, Dropout or VBLL achieve good NLL values on some datasets but underperform on others. In contrast, LEO achieves good NLL values consistently. Regarding predictive performance, LEO may underperform slightly in some cases, but consistently ranks second, whereas the method achieving the highest predictive performance varies across datasets.

Table 3: Results for tabular classification tasks. Reported values are means over 100 seeds and the values after ± are 95%-confidence intervals of the mean estimator. The best methods and z-test ties are shown in bold, and the second best methods are underlined.

| Dataset | german-credit | | | bank-marketing | | |
|---|---|---|---|---|---|---|
| Metric | Acc. (↑) | NLL (×10⁻⁴) (↓) | ECE (×10⁻²) (↓) | Acc. (↑) | NLL (×10⁻⁴) (↓) | ECE (×10⁻²) (↓) |
| Dropout | **73.77 ± 0.84** | **52.98 ± 1.47** | 10.14 ± 0.63 | 90.33 ± 0.11 | 0.46 ± 0.00 | 1.32 ± 0.08 |
| DUE | 69.72 ± 0.81 | 58.15 ± 0.67 | **9.70 ± 0.65** | 88.35 ± 0.10 | 0.74 ± 0.01 | 11.32 ± 0.21 |
| DUQ | 74.08 ± 0.89 | 51.31 ± 1.15 | 9.73 ± 0.57 | 90.14 ± 0.07 | 0.51 ± 0.00 | 2.21 ± 0.07 |
| EDL | 73.79 ± 0.83 | 52.63 ± 1.20 | 9.90 ± 0.64 | 90.39 ± 0.08 | 0.50 ± 0.00 | 3.34 ± 0.10 |
| Ensemble | 74.43 ± 0.90 | 51.19 ± 1.37 | 9.54 ± 0.57 | **90.65 ± 0.09** | **0.45 ± 0.00** | **0.93 ± 0.04** |
| VBLL | 72.64 ± 1.02 | 52.75 ± 1.22 | 9.91 ± 0.56 | **90.59 ± 0.08** | 0.46 ± 0.00 | 1.05 ± 0.07 |
| LEO (ours) | **73.99 ± 0.86** | **51.53 ± 0.88** | **8.95 ± 0.52** | 90.53 ± 0.09 | 0.46 ± 0.00 | 1.07 ± 0.05 |

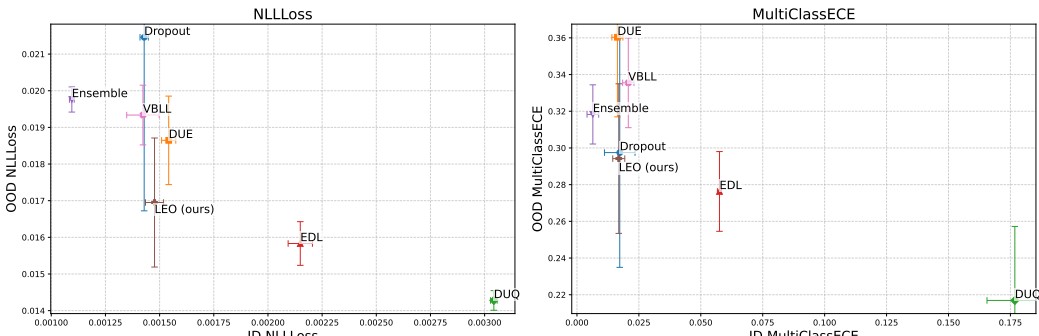

Figure 2: ID vs OoD performance on CIFAR-10 for different methods as measured by NLLLoss (left) and ECE (right). Points are means over 3 seeds and error bars correspond to 95% CIs of the mean estimator. The closer to bottom, left corner, a method is, the better.

## 4.2 CLASSIFICATION

To evaluate performance on classification tasks, we consider six tabular benchmarks (adult-census-income, bank-marketing, titanic, german-credit, breast-cancer, heart-disease) and CIFAR-10. For each benchmark, we report the negative log-likelihood (NLL, lower is better), the Expected Calibration Error (ECE, lower is better) and accuracy. On tabular benchmarks, we simply use a fully-connected architecture, whereas on CIFAR-10, we use WideResNet 28-10 as the feature extractor. To create an OoD evaluation set for CIFAR-10, we randomly corrupt the evalset images.

**Results** We present some of the results in Table 3 and Figure 2 and defer the rest to Appendices G and H. We see that LEO is able to obtain the best performance on most datasets and across most metrics, losing only in four cases (out of 18 dataset/metric combinations), in which on two of them (bank-marketing NLL and ECE) loses to Ensemble only and wins among all single-model methods. On CIFAR-10 problems, we see that most methods either excel in- or out-of-distribution, whereas LEO is able to obtain good performance in both simultaneously. This is illustrated in Figure 2, where we plot OoD performance vs ID performance according to NLL and ECE metrics (closer to the bottom-left corner indicates better performance). Ensemble excels in ID performance, but underperforms in OoD, whereas DUQ and EDL exhibit the opposite tendency. LEO, Dropout, DUE and VBLL achieve similar ID performance, but out of these four, LEO achieves the best average OoD performance, placing itself at a desirable point on the Pareto frontier.

In Table 4, we present comparison of inference times, training times and the total size of each of the models. We see LEO is one of the fastest method, having less than 1% memory higher memory footprint compared to smallest model. This is in stark contrast to Dropout, which significantly increases inference time or to Ensemble, which also significantly increases memory footprint. As such, LEO positions itself as a relatively lightweight alternative with a fast inference speed.

Table 4: Avg. inference time (with 95% CIs) and total model memory footprint for each method on CIFAR-10. Best values in bold, second best underlined.

| Metric | Baseline | Dropout | DUE | DUQ | EDL | Ensemble | VBLL | LEO (ours) |
|---|---|---|---|---|---|---|---|---|
| Infer. time (ms) (↓) | **0.06 ± 0.00** | 0.20 ± 0.00 | 0.33 ± 0.00 | **0.06 ± 0.00** | **0.06 ± 0.00** | 0.19 ± 0.00 | 0.07 ± 0.00 | **0.06 ± 0.00** |
| Model size (MB) (↓) | **139.23** | **139.23** | 156.72 | 172.34 | **139.23** | 696.13 | 154.88 | 140.89 |
| Train. time (min) (↓) | **60.80 ± 7.03** | 146.48 ± 27.77 | 79.49 ± 0.33 | 99.02 ± 34.90 | 126.28 ± 13.89 | 238.22 ± 0.97 | 76.52 ± 9.64 | **66.83 ± 7.27** |

## 4.3 BAYESIAN OPTIMISATION

Finally, we also evaluate all models used for regression experiments as surrogates for Bayesian Optimization (BO). In BO, the objective is to efficiently optimize an unknown black-box function by sequentially selecting query points. This is typically achieved by fitting a surrogate model to the observed data and then optimizing an acquisition function that balances exploration and exploitation. Crucially, the surrogate must provide reliable uncertainty estimates to enable this trade-off. Standard neural networks, which often extrapolate linearly outside the training data, tend to assign unrealis-

tically high values near the boundaries, leading the optimizer to waste queries there. By contrast, LEO defaults to a standard normal predictive distribution in extrapolation, preventing boundary regions from appearing artificially attractive. This property makes LEO particularly well-suited as a surrogate in BO.

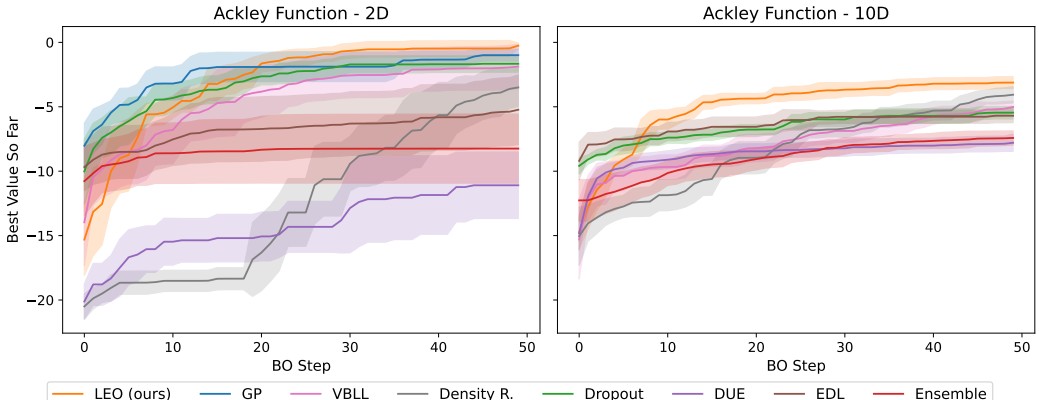

Figure 3: The best function achieved by a given optimisation step on Bayesian Optimisation tasks. Solid lines correspond to mean values over 8 seeds and shaded areas are 95% CI of the mean estimator. The objective is maximisation, thus higher values are better.

As a benchmark optimisation function, we use the popular Ackley function. We consider two problem settings with 2 and 10 dimensions, initialising the optimisation with 20 and 5000 latin hypercube samples, respectively. We use the UCB acquisition criterion with an exploration bonus of $\beta = 3$. For the 2-dimensional function, we also compare against GP, but not for the 10-dimensional cases, as fitting GP to 5000 points is infeasible. For GP, we use an RBF kernel with hyperparameters selected by optimising marginal likelihood. Optimisation curves over 50 steps are shown in Figure 3. On the 2-dimensional problem, we can see that an optimiser equipped with LEO quickly catches up with GP and even slightly outperforms it toward the end. On the 10-dimensional problem, after the first 10 steps, the optimiser using LEO clearly distinguishes itself, achieving much higher values than the other methods and maintaining the lead until the very end. These results highlight the potential of LEO for applications in sequential decision-making.

## 5 CONCLUSION

We introduced LEO, a mixture-of-experts framework that uses a distance-aware router to provide reliable epistemic uncertainty estimates. By simulating out-of-distribution scenarios during training, LEO effectively captures model uncertainty without relying on held-out OoD data. Empirical results across regression, classification, and Bayesian optimisation tasks show that LEO consistently outperforms or matches strong baselines, achieving robust performance for both in-distribution and out-of-distribution inputs. LEO can be applied to standard architectures by replacing only the last layer with a mixture of single-layer networks, introducing a negligible increase in model size and latency. These results demonstrate that LEO is a practical and scalable approach for uncertainty-aware deep learning, with promising applications in decision-making and risk-sensitive settings.

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

# A  FULL ALGORITHM PSEUDOCODE

---

**Algorithm 1** LEO Training

---

1: **Input:** Training data $\mathcal{D} = \{(x_i, y_i)\}_{i=1}^n$, types $\{t_i\}_{i=1}^n$, parameters $\psi$, $\{\theta_t\}_{t\in\mathcal{E}}$, $\phi = (\bigcup_{t\in\mathcal{E}} \phi_t) \cup \phi_0$ with $\phi_t = \{M_t, \tau_t\}$, learning rates $\eta_\psi, \eta_\theta, \eta_\phi$, epochs $E_1, E_2$
2: **Output:** Trained parameters $\psi^*, \{\theta_t^*\}_{t\in\mathcal{E}}, \phi^*$
3: Initialise $\psi, \{\theta_t\}_{t\in\mathcal{E}}, \phi$
4: **Phase 1: Expert Training**
5: **for** epoch = 1 to $E_1$ **do**
6:     **for** each mini-batch $\mathcal{B} \subset \mathcal{D}$ **do**
7:         Compute batch loss: $\mathcal{J}^{\text{experts}}(\mathcal{B}; \{\theta_t\}_{t\in\mathcal{E}}; \psi) = \frac{1}{|\mathcal{B}|} \sum_{(x_i, y_i)\in\mathcal{B}} \mathcal{L}(h_{t_i}(f(x_i; \psi); \theta_{t_i}), y_i)$
8:         Update feature extractor parameters: $\psi \leftarrow \psi - \eta_\psi \nabla_\psi \mathcal{J}^{\text{experts}}$
9:         Update expert parameters: $\theta_{t_i} \leftarrow \theta_{t_i} - \eta_\theta \nabla_{\theta_{t_i}} \mathcal{J}^{\text{experts}}$
10:     **end for**
11: **end for**
12: Freeze $\psi$ and $\{\theta_t\}_{t\in\mathcal{E}}$
13: **Phase 2: Router Training**
14: **for** epoch = 1 to $E_2$ **do**
15:     **for** each mini-batch $\mathcal{B} \subset \mathcal{D}$ **do**
16:         Sample a subset $r(\mathcal{E})$ of $\mathcal{E}$
17:         Compute router loss:

$$\mathcal{J}^{\text{router}}(\mathcal{B}; \phi) = -\frac{1}{|\mathcal{B}|} \sum_{(x_i, y_i)\in\mathcal{B}} \Big( \log p_\phi(y_i|x_i; \mathcal{E}) + \log p_\phi(y_i|x_i; \mathcal{E} \setminus (t_i \cup r(\mathcal{E}))) \Big)$$

18:         Update router parameters: $\phi \leftarrow \phi - \eta_\phi \nabla_\phi \mathcal{J}^{\text{router}}$
19:     **end for**
20: **end for**
21: **return** $\psi^*, \{\theta_t^*\}_{t\in\mathcal{E}}, \phi^*$

---

**Algorithm 2** LEO Inference (Regression)

---

1: **Input:** New input $x$, learned parameters $\psi^*, \{\theta_t^*\}_{t\in\mathcal{E}}, \phi^*$, prior mean $\mu_0(\cdot)$, prior variance $\sigma_0^2(\cdot)$
2: **Output:** Predictive mean $\hat{y}$, predictive variance $\hat{\sigma}^2$
3: Compute predictive mean:

$$\hat{y} = \mu_0(x) p_{\phi^*}(t \notin \mathcal{E}|x; \mathcal{E}) + \sum_{t\in\mathcal{E}} h_t(f(x; \psi^*); \theta_t^*) p_{\phi^*}(t|x; \mathcal{E})$$

4: Compute predictive variance:

$$\hat{\sigma}^2 = (\sigma_0^2(x) + (\mu_0(x) - \hat{y})^2) p_{\phi^*}(t \notin \mathcal{E}|x; \mathcal{E}) + \sum_{t\in\mathcal{E}} (h_t(f(x; \psi^*); \theta_t^*) - \hat{y})^2 p_{\phi^*}(t|x; \mathcal{E})$$

5: **return** $\hat{y}, \hat{\sigma}^2$

---

**Algorithm 3** LEO Inference (Classification)

---

1: **Input:** New input $x$, learned parameters $\psi^*, \{\theta_t^*\}_{t\in\mathcal{E}}, \phi^*$, prior distribution over classes $p_0(c)$
2: **Output:** Predictive categorical distribution $p(c|x)$
3: Compute prediction of each expert $t \in \mathcal{E}$ as $p(\cdot|x; t) = \text{softmax}(h_t(f(x; \psi^*)))$
4: Compute predictive probabilities for each class $c \in [C]$:

$$p(c|x) = p_0(c) p_{\phi^*}(t \notin \mathcal{E}|x; \mathcal{E}) + \sum_{t\in\mathcal{E}} p(c|x; t) p_{\phi^*}(t|x; \mathcal{E})$$

5: **return** $p(c|x)$

---

## B    DETAILS ON LEO

### B.1    PARTITIONING DATA INTO TYPES

Our training mechanism requires partitioning the training data into types before training, such that each expert sees a distinct distribution of training inputs. At the same time, we also want to create situations where one expert can make accurate predictions for at least some of the points of other types. This ensures that the router learns how much it can trust a given expert when extrapolating, which will then translate into robust uncertainty estimates for the entire model when going beyond its training domain.

To satisfy these properties, we propose to use the freshly initialised, untrained feature extractor to obtain embeddings for each point $i$, i.e., $z_i^0 = f(x_i; \psi_0)$, and then use a random projection vector $v \sim \mathcal{N}(0, \mathcal{I}_d)$ to obtain a *type indicator* $g_i = z_i^0 \cdot v$ for each data point. We then sort type indicators (which are just scalars) and split the sorted list into $|\mathcal{E}|$ consecutive chunks of equal length and give each chunk a different type in $\mathcal{E}$, which is assigned to a dedicated expert. This assignment creates mismatch across experts' training distributions, since even in a freshly initialised network, the embeddings for two data points are correlated and depend on input features in complex and random ways. At the same time, because nearby points in the sorted 1D projection are not guaranteed to be assigned to the same expert, some neighbouring points in the embedding space may be split across experts, allowing partial extrapolation and forcing the router to learn expert reliability.

### B.2    RANDOM SUBSET SELECTION

To obtain the random subset of experts to drop for a given data point $i$, we first sample $u_i \sim U(0, 1)$ and then for each expert associated with $t \in \mathcal{E}$, we sample $m_{t,i} \sim U(0, 1)$. We drop the $t$th expert for the $i$th data point if $m_{t,i} < u_i$. In this way, we drop experts with uniform probability, but also the average number of experts we drop is uniformly distributed. We do this with the objective of making the model more robust by simulating more diverse OoD scenarios.

### B.3    UPDATING CENTROIDS

We utilise the same moving-average-style update rule for the centroids $e_t$ as the one employed in DUQ (Van Amersfoort et al., 2020), i.e., after each mini-batch $\{(x_i, t_i, y_i)\}_{i=1}^{|\mathcal{B}|}$ of size $|\mathcal{B}|$ we update

$$N_t := \gamma\, N_t + (1 - \gamma)\, n_t, \tag{3}$$

$$m_t := \gamma\, m_t + (1 - \gamma)\, \frac{1}{n_t} \sum_{\{i : t_i = t\}} f(x_i; \psi)^T M_t, \tag{4}$$

$$e_t := \frac{m_t}{N_t}. \tag{5}$$

where $n_t = |\{i : t_i = t\}|$. We initialise $N_t = 13$ for all types and initalise $m_t$ with small Gaussian noise $\mathcal{N}(0, 0.05^2)$. We set $\gamma = 0.99$.

## C    DETAILED EXPERIMENTAL SETUP

For our experiments, we used machines with NVIDIA A40 GPUs with 48 GB of memory.

### C.1    GENERAL

We try to keep the experimental setup as similar as possible across methods. For this reason, across all experiments, we use 5 models for Ensemble, 5 dropout samples for Dropout, and 5 experts for LEO. We use a dropout rate of 0.3 for the Dropout method. Across all methods, we keep the architecture fixed except for the last layer, which changes depending on the exact method used (e.g. Variational GP in DUE or expert heads and router in LEO). In VBLL, we use the same optimiser setting as the original authors, namely we use a weight decay of 0.01 and clip max gradient to 1.0

across all experiments, and thus use AdamW, while other baselines use Adam. For DUE, we use the RBF kernel and set the number of inducing points equal to the number of classes in classification and use 20 inducing points in regression. For baselines requiring a distance-preserving feature extractor (DUE, DUQ), we apply spectral normalization to the feature extractor and add residual connections if they are not present by default (e.g. when the feature extractor is just a fully-connected network). For EDL, performance is highly sensitive to the hyperparmeter $\lambda$. We tune the $\lambda$ hyperparameter by first running 20% of the total training iterations with different values of $\lambda$ and choosing the one that produces the best validation likelihood at the end of training. For LEO, we use the same number of epochs as other methods to train the experts and then the same number of epochs to train the router (which is much faster, as experts and feature extractor are fixed). In all experiments across all baselines we use a "patience" mechanism, i.e., if the last epoch achieved the best validation loss, we extend training until the validation loss stops improving.

## C.2 REGRESSION - UCI

For each problem and baseline, we use a fully-connected network with three hidden layers of size 256 with relu nonlinearities. We train for a total of 10000 epochs with Adam with the learning rate set to 0.001. We use full-batch gradient descent. We measure the NLL on the validation set every 100 epochs and select the checkpoint with the lowest value.

## C.3 REGRESSION - UTK

We use freshly initalised ResNet-18, followed by one fully-connected layer. We train for a total of 50 epochs with a batch size of 128 using Adam with a learning rate equal to 0.00001. We measure the NLL on the validation set after every epoch and select the checkpoint with the lowest value.

## C.4 CLASSIFICATION - TABULAR

For each problem and baseline, we use a full-connected network with two hidden layers of size 256 with relu nonlinearities. We use Adam with a learning rate of 0.01 and full batch gradient descent. We measure the NLL on the validation set after every epoch and select the checkpoint with the lowest value.

## C.5 CLASSIFICATION - CIFAR-10

We use freshly initialised WideResNet 28-10 with a dropout rate of 0.3, followed by one fully-connected layer, outputting logits for each of the ten classes. We use the same data augmentation as in Zagoruyko & Komodakis (2016). We train for 50 epochs using SGD with momentum equal to 0.9 and weight decay set to $5 \times 10^{-4}$. We start with a learning rate of 0.1 and divide it by 5 after 20, 30 and 40 epochs. We use a batch size of 128.

## C.6 BAYESIAN OPTIMISATION

We use the same architecture and training setup as in Regression - UCI. However, since in BO we need to be extremely sample efficient, we use all the availalbe data points for training and do not reserve a validation set. Instead, we use weight decay of 0.01 and therefore the AdamW optimiser. Given the model predicts a mean $\mu(x)$ and variance $\sigma^2(x)$ at a given point $x$, we select the next point to query by maximising the UCB acquistion function $\alpha(x) = \mu(x) + \beta\sigma(x)$ and set $\beta = 3$. We use BoTorch (Balandat et al., 2020) to optimise the acquisition function. We completely retrain each model after acquiring a new point.

## D  OBTAINING OOD EVALUATION SETS IN REGRESSION TASKS

On the UCI benchmarks, eight datasets have only in-distribution evaluation sets, whereas two of them (protein and wine) have both in- and out-of-distribution evaluation sets. For the wine dataset, we follow Bui & Liu (2024), using red wines for training and ID evaluation and white wines for OoD evaluation. For the protein dataset, we follow Ziomek et al. (2025), using smaller proteins

for training and ID evaluation and larger proteins for OoD evaluation. In each case, we simply use fully-connected architectures. On UTK-Face, to create ID and OoD evaluation sets, we follow Ziomek et al. (2025), using all images with ethnicity label "Others" as the OoD evaluation set and all remaining ethnicities as training and ID evaluation sets. We use freshly initialised ResNet-18 as the backbone model.

# E   OBTAINING OOD EVALUATION SETS IN CIFAR-10

To evaluate OoD robustness, we construct a corrupted variant of CIFAR by applying common image corruptions. For each image, one corruption type is chosen at random. The set of corruption types includes:

- **Gaussian noise:** additive pixel-wise Gaussian noise

- **Salt-and-pepper noise:** randomly setting pixels to black or white

- **Gaussian blur:** convolution with a Gaussian kernel

- **Motion blur:** convolution with a horizontal motion

- **Brightness shift:** multiplicative rescaling of pixel intensities by a random factor

- **Contrast reduction:** pixel intensities are shifted toward the per-image mean

- **Pixelation:** downsampling the image followed by nearest-neighbor upsampling.

All corrupted images are clipped to the valid pixel range $[0, 255]$.

# F   DETAILED UCI RESULTS

Table 5: Results for UCI benchmarks. Reported values are means over 20 seeds and the values after $\pm$ denote 95% CIs of the mean estimator. The best methods and all methods that do not statistically differ w.r.t. two-sided z-test are shown in bold. The second best methods are underlined.

| Dataset | boston | | california | | concrete | | energy-efficiency | |
|---|---|---|---|---|---|---|---|---|
| Metric | NLL ($\downarrow$) | R2 ($\uparrow$) | NLL ($\downarrow$) | R2 ($\uparrow$) | NLL ($\downarrow$) | R2 ($\uparrow$) | NLL ($\downarrow$) | R2 ($\uparrow$) |
| Density R. | $\underline{0.98 \pm 0.56}$ | $0.81 \pm 0.05$ | $0.55 \pm 0.05$ | $0.76 \pm 0.04$ | $0.64 \pm 0.17$ | $0.89 \pm 0.01$ | $1.46 \pm 0.86$ | $0.98 \pm 0.00$ |
| Dropout | $3.71 \pm 0.72$ | $\mathbf{0.87 \pm 0.02}$ | $3.91 \pm 0.26$ | $\mathbf{0.82 \pm 0.01}$ | $3.06 \pm 0.96$ | $\mathbf{0.91 \pm 0.01}$ | $\mathbf{-0.74 \pm 0.20}$ | $\mathbf{0.99 \pm 0.00}$ |
| DUE | $2.30 \pm 0.40$ | $0.54 \pm 0.10$ | $2.64 \pm 0.14$ | $0.31 \pm 0.07$ | $2.08 \pm 0.15$ | $0.55 \pm 0.09$ | $\mathbf{-0.79 \pm 0.17}$ | $\mathbf{0.99 \pm 0.00}$ |
| EDL | $\mathbf{0.47 \pm 0.18}$ | $\mathbf{0.86 \pm 0.02}$ | $\mathbf{0.49 \pm 0.05}$ | $\mathbf{0.48 \pm 0.57}$ | $\mathbf{0.36 \pm 0.15}$ | $\mathbf{0.90 \pm 0.01}$ | $\mathbf{-0.91 \pm 0.10}$ | $0.98 \pm 0.01$ |
| Ensemble | $8.26 \pm 3.18$ | $\mathbf{0.87 \pm 0.02}$ | $2.33 \pm 0.14$ | $0.77 \pm 0.01$ | $4.75 \pm 1.32$ | $\mathbf{0.92 \pm 0.01}$ | $\underline{0.37 \pm 0.46}$ | $\mathbf{0.99 \pm 0.00}$ |
| VBLL | $\mathbf{3.23 \pm 5.21}$ | $\mathbf{0.86 \pm 0.02}$ | $1.33 \pm 0.41$ | $0.71 \pm 0.02$ | $4.51 \pm 3.83$ | $\mathbf{0.90 \pm 0.02}$ | $\underline{1.74 \pm 1.90}$ | $0.98 \pm 0.00$ |
| LEO (ours) | $\mathbf{0.35 \pm 0.14}$ | $\underline{0.84 \pm 0.02}$ | $\mathbf{0.45 \pm 0.03}$ | $\underline{0.79 \pm 0.01}$ | $\mathbf{0.29 \pm 0.12}$ | $\underline{0.89 \pm 0.01}$ | $\mathbf{-0.87 \pm 0.08}$ | $0.98 \pm 0.00$ |

Table 6: Results for the wine UCI benchmark. The training set and the ID evaluation set correspond to red wine. Reported values are means over 20 seeds and the values after $\pm$ denote 95% CIs of the mean estimator. The best methods and all methods that do not statistically differ w.r.t. two-sided z-test are shown in bold. The second best methods are underlined. In cells marked with (*), OOD evaluation for Density Regression caused numerical instabilities on 2 out of 20 seeds, omitting those cases average value reached were 3.23 for OOD NLL and $-8.36 \times 10^{23}$ for OOD $R^2$. In cell marked with (†), OOD NLL evaluation for EDL caused numerical instabilities on 7 out of 20 seeds, remaining seeds reached an average OOD NLL equal to 9051.13.

| Dataset | wine | | | |
|---|---|---|---|---|
| Metric | NLL ($\downarrow$) | R2 ($\uparrow$) | OOD NLL ($\downarrow$) | OOD R2 ($\uparrow$) |
| Density R. | $1.54 \pm 0.06$ | $\mathbf{-4.00 \pm 5.51}$ | (*) | (*) |
| Dropout | $10.51 \pm 3.19$ | $\mathbf{0.40 \pm 0.02}$ | $12.04 \pm 2.12$ | $\mathbf{0.07 \pm 0.04}$ |
| DUE | $4.38 \pm 0.25$ | $0.11 \pm 0.01$ | $6.12 \pm 0.26$ | $\underline{-0.01 \pm 0.01}$ |
| EDL | $\mathbf{2.76 \pm 1.59}$ | $\underline{0.36 \pm 0.02}$ | (†) | $\underline{-0.04 \pm 0.06}$ |
| Ensemble | $5.70 \pm 0.79$ | $0.32 \pm 0.03$ | $4.95 \pm 0.46$ | $-0.27 \pm 0.08$ |
| VBLL | $112.55 \pm 105.56$ | $0.29 \pm 0.04$ | $136.16 \pm 82.00$ | $-0.39 \pm 0.10$ |
| LEO (ours) | $\mathbf{1.23 \pm 0.03}$ | $\underline{0.37 \pm 0.02}$ | $\mathbf{1.55 \pm 0.03}$ | $\underline{-0.02 \pm 0.05}$ |

## G  DETAILED TABULAR CLASSIFICATION RESULTS

Table 7: Results for tabular classification tasks. Reported values are means over 100 seeds and the values after $\pm$ are 95%-confidence intervals of the mean estimator. The best methods and z-test ties are shown in bold, and the second best methods are underlined.

| Dataset | adult | | | breast-cancer | | |
|---|---|---|---|---|---|---|
| Metric | Acc. ($\uparrow$) | NLL ($\times10^{-4}$) ($\downarrow$) | ECE ($\times10^{-2}$) ($\downarrow$) | Acc. ($\uparrow$) | NLL ($\times10^{-4}$) ($\downarrow$) | ECE ($\times10^{-2}$) ($\downarrow$) |
| Dropout | $\mathbf{85.74 \pm 0.09}$ | $\mathbf{0.63 \pm 0.00}$ | $\mathbf{1.08 \pm 0.06}$ | $\mathbf{96.67 \pm 0.46}$ | $\underline{18.49 \pm 3.07}$ | $\mathbf{3.54 \pm 0.34}$ |
| DUE | $76.22 \pm 0.11$ | $0.99 \pm 0.01$ | $13.46 \pm 0.29$ | $69.75 \pm 1.66$ | $80.04 \pm 2.42$ | $18.94 \pm 1.51$ |
| DUQ | $84.81 \pm 0.10$ | $0.67 \pm 0.00$ | $1.82 \pm 0.08$ | $\mathbf{96.06 \pm 0.49}$ | $18.34 \pm 2.21$ | $4.21 \pm 0.32$ |
| EDL | $\underline{85.50 \pm 0.26}$ | $0.65 \pm 0.00$ | $2.74 \pm 0.23$ | $\mathbf{96.70 \pm 0.42}$ | $17.06 \pm 1.90$ | $\mathbf{3.56 \pm 0.35}$ |
| Ensemble | $\mathbf{85.78 \pm 0.09}$ | $\mathbf{0.63 \pm 0.00}$ | $1.04 \pm 0.05$ | $\mathbf{96.71 \pm 0.40}$ | $18.08 \pm 2.74$ | $\mathbf{3.45 \pm 0.28}$ |
| VBLL | $\underline{85.61 \pm 0.10}$ | $\underline{0.64 \pm 0.00}$ | $1.12 \pm 0.06$ | $\mathbf{96.39 \pm 0.51}$ | $\mathbf{14.82 \pm 1.67}$ | $\underline{4.18 \pm 0.39}$ |
| LEO (ours) | $\mathbf{85.77 \pm 0.09}$ | $\mathbf{0.63 \pm 0.00}$ | $\underline{1.25 \pm 0.07}$ | $\mathbf{96.71 \pm 0.40}$ | $16.66 \pm 1.34$ | $4.69 \pm 0.30$ |

Table 8: Results for tabular classification tasks. Reported values are means over 100 seeds and the values after $\pm$ are 95%-confidence intervals of the mean estimator. The best methods and z-test ties are shown in bold, and the second best methods are underlined.

| Dataset | heart-disease | | | titanic | | |
|---|---|---|---|---|---|---|
| Metric | Acc. ($\uparrow$) | NLL ($\times10^{-4}$) ($\downarrow$) | ECE ($\times10^{-2}$) ($\downarrow$) | Acc. ($\uparrow$) | NLL ($\times10^{-4}$) ($\downarrow$) | ECE ($\times10^{-2}$) ($\downarrow$) |
| Dropout | $\mathbf{82.07 \pm 1.42}$ | $209.74 \pm 35.29$ | $\mathbf{15.78 \pm 1.04}$ | $\mathbf{79.75 \pm 0.65}$ | $\mathbf{35.48 \pm 0.83}$ | $\mathbf{7.83 \pm 0.42}$ |
| DUE | $74.85 \pm 1.96$ | $219.75 \pm 3.45$ | $\underline{20.44 \pm 1.29}$ | $62.12 \pm 1.34$ | $\underline{50.63 \pm 0.71}$ | $\underline{8.33 \pm 0.61}$ |
| DUQ | $\mathbf{83.48 \pm 1.40}$ | $\mathbf{145.89 \pm 8.27}$ | $14.87 \pm 0.80$ | $\underline{78.95 \pm 0.71}$ | $\mathbf{35.40 \pm 0.73}$ | $\mathbf{7.83 \pm 0.41}$ |
| EDL | $\mathbf{82.93 \pm 1.45}$ | $165.60 \pm 13.39$ | $15.22 \pm 0.99$ | $\underline{79.54 \pm 0.68}$ | $36.31 \pm 0.91$ | $8.08 \pm 0.47$ |
| Ensemble | $\underline{81.37 \pm 1.52}$ | $\underline{187.97 \pm 33.33}$ | $15.15 \pm 1.00$ | $\mathbf{79.98 \pm 0.69}$ | $35.53 \pm 0.93$ | $\mathbf{7.64 \pm 0.43}$ |
| VBLL | $\underline{80.56 \pm 1.71}$ | $\underline{167.01 \pm 10.62}$ | $15.99 \pm 1.08$ | $\mathbf{79.45 \pm 0.94}$ | $35.96 \pm 1.01$ | $8.07 \pm 0.45$ |
| LEO (ours) | $\mathbf{82.81 \pm 1.45}$ | $\mathbf{153.60 \pm 6.56}$ | $15.75 \pm 0.76$ | $\mathbf{79.43 \pm 0.73}$ | $35.71 \pm 0.76$ | $\mathbf{7.60 \pm 0.39}$ |

# H FULL CIFAR-10 RESULTS

Table 9: Results for CIFAR-10 benchmarks. Reported values are means over 3 seeds and the values after $\pm$ denote 95% CIs of the mean estimator. The best methods and all methods that do not statistically differ w.r.t. two-sided z-test are shown in bold. The second best methods are underlined.

| Dataset | ID | | | OOD | | |
|---|---|---|---|---|---|---|
| Metric | Acc ($\uparrow$) | NLLLoss ($\times 10^{-3}$) ($\downarrow$) | ECE ($\times 10^{-3}$) ($\downarrow$) | Acc ($\uparrow$) | NLLLoss ($\times 10^{-3}$) ($\downarrow$) | ECE ($\times 10^{-3}$) ($\downarrow$) |
| Dropout | $94.44 \pm 0.47$ | $1.43 \pm 0.02$ | $17.27 \pm 6.17$ | $\mathbf{44.79 \pm 1.17}$ | $21.46 \pm 4.74$ | $\underline{297.47 \pm 62.55}$ |
| DUE | $\underline{94.54 \pm 0.16}$ | $1.54 \pm 0.03$ | $\underline{16.33 \pm 2.23}$ | $\mathbf{44.52 \pm 1.43}$ | $18.65 \pm 1.21$ | $360.31 \pm 43.37$ |
| DUQ | $93.75 \pm 0.39$ | $3.04 \pm 0.02$ | $176.79 \pm 11.24$ | $\underline{44.14 \pm 1.23}$ | $\mathbf{14.28 \pm 0.27}$ | $\mathbf{216.90 \pm 40.33}$ |
| EDL | $93.99 \pm 0.40$ | $2.15 \pm 0.06$ | $57.54 \pm 0.83$ | $42.72 \pm 1.38$ | $\underline{15.83 \pm 0.60}$ | $276.30 \pm 21.73$ |
| Ensemble | $\mathbf{95.44 \pm 0.08}$ | $\mathbf{1.09 \pm 0.01}$ | $\mathbf{6.42 \pm 2.35}$ | $\mathbf{45.58 \pm 0.35}$ | $19.76 \pm 0.34$ | $318.27 \pm 16.13$ |
| VBLL | $\underline{94.72 \pm 0.54}$ | $\underline{1.42 \pm 0.07}$ | $20.73 \pm 2.29$ | $\mathbf{44.23 \pm 1.50}$ | $19.34 \pm 0.81$ | $335.49 \pm 24.44$ |
| LEO (ours) | $94.17 \pm 0.50$ | $1.48 \pm 0.04$ | $16.88 \pm 2.43$ | $44.18 \pm 0.12$ | $16.95 \pm 1.76$ | $294.20 \pm 40.77$ |

# I   ABLATIONS ON NUMBER OF EXPERTS

Table 10: Ablations results for UCI benchmarks. We compare a version of LEO with 5 experts that we used throughout experiments to versions of LEO with 2,3 and 10 experts. Reported values are means over 20 seeds and the values after ± denote 95% CIs of the mean estimator. The best methods and all methods that do not statistically differ w.r.t. two-sided z-test are shown in bold. The second best methods are underlined. See Table 6 for explanation of ($\star$) and ($\dagger$) symbols.

| Dataset | boston | | california | | concrete | | energy-efficiency | |
| Metric | NLL ($\downarrow$) | R2 ($\uparrow$) | NLL ($\downarrow$) | R2 ($\uparrow$) | NLL ($\downarrow$) | R2 ($\uparrow$) | NLL ($\downarrow$) | R2 ($\uparrow$) |
|---|---|---|---|---|---|---|---|---|
| Density R. | $0.99 \pm 0.53$ | $0.81 \pm 0.04$ | $0.55 \pm 0.05$ | $0.76 \pm 0.04$ | $0.62 \pm 0.16$ | $\underline{0.89 \pm 0.01}$ | $1.96 \pm 1.28$ | $0.98 \pm 0.00$ |
| Dropout | $3.75 \pm 0.69$ | $\mathbf{0.87 \pm 0.02}$ | $3.90 \pm 0.25$ | $\mathbf{0.82 \pm 0.01}$ | $2.95 \pm 0.93$ | $\mathbf{0.91 \pm 0.01}$ | $\underline{-0.75 \pm 0.19}$ | $\mathbf{0.99 \pm 0.00}$ |
| DUE | $2.28 \pm 0.38$ | $0.55 \pm 0.10$ | $2.65 \pm 0.13$ | $0.31 \pm 0.07$ | $2.07 \pm 0.15$ | $0.55 \pm 0.09$ | $-0.80 \pm 0.17$ | $\mathbf{0.99 \pm 0.00}$ |
| EDL | $\underline{0.46 \pm 0.17}$ | $\mathbf{0.86 \pm 0.02}$ | $0.49 \pm 0.05$ | $0.51 \pm 0.53$ | $0.36 \pm 0.14$ | $0.90 \pm 0.01$ | $\mathbf{-0.91 \pm 0.09}$ | $0.98 \pm 0.01$ |
| Ensemble | $8.02 \pm 3.06$ | $\mathbf{0.87 \pm 0.02}$ | $2.33 \pm 0.13$ | $0.77 \pm 0.01$ | $4.66 \pm 1.26$ | $\mathbf{0.92 \pm 0.01}$ | $0.34 \pm 0.44$ | $\mathbf{0.99 \pm 0.00}$ |
| VBLL | $3.09 \pm 4.97$ | $\mathbf{0.87 \pm 0.02}$ | $1.31 \pm 0.39$ | $0.71 \pm 0.02$ | $4.31 \pm 3.67$ | $\underline{0.90 \pm 0.02}$ | $1.75 \pm 1.81$ | $0.98 \pm 0.00$ |
| LEO (ours) (2 experts) | $\mathbf{0.29 \pm 0.06}$ | $\underline{0.83 \pm 0.02}$ | $0.47 \pm 0.02$ | $0.79 \pm 0.01$ | $\mathbf{0.28 \pm 0.12}$ | $0.90 \pm 0.01$ | $\mathbf{-0.84 \pm 0.08}$ | $0.98 \pm 0.00$ |
| LEO (ours) (3 experts) | $\mathbf{0.30 \pm 0.07}$ | $0.84 \pm 0.02$ | $0.47 \pm 0.02$ | $0.79 \pm 0.01$ | $0.31 \pm 0.14$ | $0.90 \pm 0.01$ | $-0.78 \pm 0.13$ | $0.98 \pm 0.00$ |
| LEO (ours) (5 experts) | $\underline{0.37 \pm 0.16}$ | $0.84 \pm 0.02$ | $0.46 \pm 0.03$ | $0.79 \pm 0.01$ | $\mathbf{0.23 \pm 0.08}$ | $0.89 \pm 0.01$ | $\mathbf{-0.86 \pm 0.08}$ | $0.98 \pm 0.00$ |
| LEO (ours) (10 experts) | $\mathbf{0.28 \pm 0.06}$ | $0.83 \pm 0.02$ | $\mathbf{0.43 \pm 0.02}$ | $0.79 \pm 0.01$ | $0.25 \pm 0.09$ | $0.89 \pm 0.01$ | $-0.63 \pm 0.10$ | $0.97 \pm 0.00$ |

| Dataset | kin8nm | | naval | | power-plant | | yacht | |
| Metric | NLL ($\downarrow$) | R2 ($\uparrow$) | NLL ($\downarrow$) | R2 ($\uparrow$) | NLL ($\downarrow$) | R2 ($\uparrow$) | NLL ($\downarrow$) | R2 ($\uparrow$) |
|---|---|---|---|---|---|---|---|---|
| Density R. | $0.19 \pm 0.03$ | $\underline{0.92 \pm 0.00}$ | $-2.24 \pm 0.05$ | $\mathbf{1.00 \pm 0.00}$ | $\mathbf{-0.09 \pm 0.02}$ | $0.95 \pm 0.00$ | $1.17 \pm 1.15$ | $0.99 \pm 0.00$ |
| Dropout | $1.18 \pm 0.12$ | $\underline{0.92 \pm 0.00}$ | $-1.12 \pm 0.02$ | $0.99 \pm 0.00$ | $3.10 \pm 0.33$ | $0.95 \pm 0.00$ | $-1.20 \pm 0.26$ | $0.98 \pm 0.01$ |
| DUE | $1.96 \pm 0.12$ | $0.79 \pm 0.02$ | $-0.42 \pm 0.31$ | $\mathbf{1.00 \pm 0.00}$ | $1.19 \pm 0.09$ | $0.89 \pm 0.00$ | $-1.49 \pm 0.05$ | $\mathbf{1.00 \pm 0.00}$ |
| EDL | $0.18 \pm 0.03$ | $0.91 \pm 0.01$ | $-1.84 \pm 0.02$ | $\mathbf{1.00 \pm 0.00}$ | $\mathbf{-0.09 \pm 0.03}$ | $0.95 \pm 0.00$ | $\underline{-2.08 \pm 0.33}$ | $0.99 \pm 0.00$ |
| Ensemble | $1.29 \pm 0.19$ | $\mathbf{0.93 \pm 0.00}$ | $-2.27 \pm 0.04$ | $\mathbf{1.00 \pm 0.00}$ | $1.68 \pm 0.24$ | $\mathbf{0.96 \pm 0.00}$ | $\underline{-2.52 \pm 0.31}$ | $\mathbf{1.00 \pm 0.00}$ |
| VBLL | $2.64 \pm 1.91$ | $0.89 \pm 0.00$ | $-0.54 \pm 0.24$ | $0.99 \pm 0.00$ | $\underline{-0.04 \pm 0.03}$ | $0.95 \pm 0.00$ | $0.06 \pm 0.89$ | $0.99 \pm 0.00$ |
| LEO (ours) (2 experts) | $0.16 \pm 0.02$ | $\underline{0.92 \pm 0.00}$ | $\mathbf{-2.58 \pm 0.08}$ | $\mathbf{1.00 \pm 0.00}$ | $-0.00 \pm 0.04$ | $0.95 \pm 0.00$ | $\underline{-2.15 \pm 0.28}$ | $\mathbf{1.00 \pm 0.00}$ |
| LEO (ours) (3 experts) | $0.16 \pm 0.01$ | $\underline{0.92 \pm 0.00}$ | $\underline{-2.46 \pm 0.07}$ | $\mathbf{1.00 \pm 0.00}$ | $\underline{-0.05 \pm 0.03}$ | $0.95 \pm 0.00$ | $\underline{-1.89 \pm 0.59}$ | $\mathbf{1.00 \pm 0.00}$ |
| LEO (ours) (5 experts) | $0.14 \pm 0.01$ | $\underline{0.92 \pm 0.00}$ | $\mathbf{-2.58 \pm 0.08}$ | $\mathbf{1.00 \pm 0.00}$ | $\mathbf{-0.06 \pm 0.03}$ | $\underline{0.95 \pm 0.00}$ | $-2.22 \pm 0.17$ | $0.99 \pm 0.00$ |
| LEO (ours) (10 experts) | $\mathbf{0.11 \pm 0.01}$ | $\underline{0.92 \pm 0.00}$ | $\mathbf{-2.64 \pm 0.08}$ | $\mathbf{1.00 \pm 0.00}$ | $\underline{-0.05 \pm 0.05}$ | $0.95 \pm 0.00$ | $-1.66 \pm 0.28$ | $\underline{0.99 \pm 0.00}$ |

| Dataset | protein | | | | wine | | | |
| Metric | NLL ($\downarrow$) | R2 ($\uparrow$) | OOD NLL ($\downarrow$) | OOD R2 ($\uparrow$) | NLL ($\downarrow$) | R2 ($\uparrow$) | OOD NLL ($\downarrow$) | OOD R2 ($\uparrow$) |
|---|---|---|---|---|---|---|---|---|
| Density R. | $1.00 \pm 0.23$ | (*) | $11.64 \pm 2.38$ | $0.39 \pm 0.05$ | $1.54 \pm 0.06$ | $-3.83 \pm 5.25$ | $3.27 \pm 0.46$ | (*) |
| Dropout | $4.38 \pm 0.53$ | $\mathbf{0.69 \pm 0.00}$ | $4.82 \pm 0.39$ | $\mathbf{0.54 \pm 0.01}$ | $10.52 \pm 3.03$ | $\mathbf{0.40 \pm 0.02}$ | $12.15 \pm 2.03$ | $\mathbf{0.07 \pm 0.04}$ |
| DUE | $5.12 \pm 0.20$ | $0.08 \pm 0.01$ | $2.99 \pm 0.11$ | $0.14 \pm 0.01$ | $4.35 \pm 0.25$ | $0.11 \pm 0.01$ | $6.10 \pm 0.25$ | $-0.01 \pm 0.01$ |
| EDL | $1.07 \pm 0.02$ | $0.41 \pm 0.01$ | $\mathbf{1.16 \pm 0.07}$ | $0.44 \pm 0.03$ | $2.70 \pm 1.51$ | $\underline{0.36 \pm 0.02}$ | ($\dagger$) | $-0.04 \pm 0.06$ |
| Ensemble | $2.27 \pm 0.11$ | $\underline{0.68 \pm 0.00}$ | $1.89 \pm 0.20$ | $0.27 \pm 0.04$ | $5.66 \pm 0.76$ | $0.32 \pm 0.03$ | $4.90 \pm 0.46$ | $-0.28 \pm 0.08$ |
| VBLL | $1.02 \pm 0.04$ | $\underline{0.59 \pm 0.01}$ | $2.46 \pm 0.45$ | $-0.20 \pm 0.14$ | $107.26 \pm 100.94$ | $0.29 \pm 0.04$ | $129.77 \pm 78.99$ | $-0.39 \pm 0.10$ |
| LEO (ours) (2 experts) | $0.95 \pm 0.04$ | $0.61 \pm 0.00$ | $\mathbf{1.17 \pm 0.04}$ | $\underline{0.44 \pm 0.02}$ | $\underline{1.26 \pm 0.02}$ | $0.33 \pm 0.01$ | $\mathbf{1.54 \pm 0.02}$ | $\underline{0.02 \pm 0.04}$ |
| LEO (ours) (3 experts) | $\underline{0.91 \pm 0.04}$ | $0.61 \pm 0.01$ | $\mathbf{1.16 \pm 0.04}$ | $\underline{0.43 \pm 0.02}$ | $\underline{1.24 \pm 0.03}$ | $0.35 \pm 0.02$ | $\mathbf{1.53 \pm 0.03}$ | $\underline{0.03 \pm 0.04}$ |
| LEO (ours) (5 experts) | $\mathbf{0.89 \pm 0.03}$ | $0.60 \pm 0.00$ | $\mathbf{1.16 \pm 0.05}$ | $\underline{0.44 \pm 0.02}$ | $\mathbf{1.23 \pm 0.03}$ | $0.37 \pm 0.02$ | $1.56 \pm 0.03$ | $-0.03 \pm 0.04$ |
| LEO (ours) (10 experts) | $\mathbf{0.86 \pm 0.05}$ | $0.59 \pm 0.01$ | $\mathbf{1.16 \pm 0.04}$ | $0.39 \pm 0.03$ | $\mathbf{1.20 \pm 0.03}$ | $\mathbf{0.40 \pm 0.02}$ | $\underline{1.55 \pm 0.02}$ | $-0.03 \pm 0.04$ |

## J  ABLATION ON TYPE ASSIGNMENT

Table 11: Ablations results for UCI benchmarks. We compare different ways of assigning expert types. "Original" refers to the method we use throughout other experiments described in B.1. "KMeans" refers to a variation of that method, where instead of constructing histogram we conduct Kmeans clustering and "Random" is just a purely random type assignment. Reported values are means over 20 seeds and the values after $\pm$ denote 95% CIs of the mean estimator. The best methods and all methods that do not statistically differ w.r.t. two-sided z-test are shown in bold. The second best methods are underlined. See Table 6 for explanation of $(\star)$ and $(\dagger)$ symbols.

| Dataset | boston | | california | | concrete | | energy-efficiency | |
|---|---|---|---|---|---|---|---|---|
| Metric | NLL ($\downarrow$) | R2 ($\uparrow$) | NLL ($\downarrow$) | R2 ($\uparrow$) | NLL ($\downarrow$) | R2 ($\uparrow$) | NLL ($\downarrow$) | R2 ($\uparrow$) |
| Density R. | $0.99 \pm 0.53$ | $0.81 \pm 0.04$ | $0.55 \pm 0.05$ | $0.76 \pm 0.04$ | $0.62 \pm 0.16$ | $0.89 \pm 0.01$ | $1.96 \pm 1.28$ | $0.98 \pm 0.00$ |
| Dropout | $3.75 \pm 0.69$ | $\mathbf{0.87 \pm 0.02}$ | $3.90 \pm 0.25$ | $\mathbf{0.82 \pm 0.01}$ | $2.95 \pm 0.93$ | $\mathbf{0.91 \pm 0.01}$ | $\underline{-0.75 \pm 0.19}$ | $\mathbf{0.99 \pm 0.00}$ |
| DUE | $2.28 \pm 0.38$ | $0.55 \pm 0.10$ | $2.65 \pm 0.13$ | $0.31 \pm 0.07$ | $2.07 \pm 0.15$ | $0.55 \pm 0.09$ | $\underline{-0.80 \pm 0.17}$ | $\mathbf{0.99 \pm 0.00}$ |
| EDL | $\underline{0.46 \pm 0.17}$ | $\mathbf{0.86 \pm 0.02}$ | $\underline{0.49 \pm 0.05}$ | $0.51 \pm 0.53$ | $\underline{0.36 \pm 0.14}$ | $0.90 \pm 0.01$ | $\mathbf{-0.91 \pm 0.09}$ | $0.98 \pm 0.01$ |
| Ensemble | $8.02 \pm 3.06$ | $\mathbf{0.87 \pm 0.02}$ | $2.33 \pm 0.13$ | $0.77 \pm 0.01$ | $4.66 \pm 1.26$ | $\mathbf{0.92 \pm 0.01}$ | $0.34 \pm 0.44$ | $\mathbf{0.99 \pm 0.00}$ |
| VBLL | $3.09 \pm 4.97$ | $\mathbf{0.87 \pm 0.02}$ | $1.31 \pm 0.39$ | $0.71 \pm 0.02$ | $4.31 \pm 3.67$ | $0.90 \pm 0.02$ | $1.75 \pm 1.81$ | $0.98 \pm 0.00$ |
| LEO (Original) | $\mathbf{0.37 \pm 0.16}$ | $\underline{0.84 \pm 0.02}$ | $\mathbf{0.46 \pm 0.03}$ | $0.79 \pm 0.01$ | $\mathbf{0.23 \pm 0.08}$ | $0.89 \pm 0.01$ | $-0.86 \pm 0.08$ | $0.98 \pm 0.00$ |
| LEO (KMeans) | $\mathbf{0.27 \pm 0.08}$ | $\underline{0.83 \pm 0.02}$ | $\mathbf{0.47 \pm 0.02}$ | $0.78 \pm 0.01$ | $\mathbf{0.26 \pm 0.10}$ | $0.89 \pm 0.01$ | $-0.69 \pm 0.13$ | $0.97 \pm 0.01$ |
| LEO (Random) | $\mathbf{0.24 \pm 0.05}$ | $\underline{0.84 \pm 0.02}$ | $\mathbf{0.45 \pm 0.02}$ | $0.79 \pm 0.01$ | $\mathbf{0.22 \pm 0.08}$ | $0.89 \pm 0.01$ | $-0.81 \pm 0.08$ | $0.98 \pm 0.00$ |

| Dataset | kin8nm | | naval | | power-plant | | yacht | |
|---|---|---|---|---|---|---|---|---|
| Metric | NLL ($\downarrow$) | R2 ($\uparrow$) | NLL ($\downarrow$) | R2 ($\uparrow$) | NLL ($\downarrow$) | R2 ($\uparrow$) | NLL ($\downarrow$) | R2 ($\uparrow$) |
| Density R. | $0.19 \pm 0.03$ | $\underline{0.92 \pm 0.00}$ | $-2.24 \pm 0.05$ | $\mathbf{1.00 \pm 0.00}$ | $\mathbf{-0.09 \pm 0.02}$ | $0.95 \pm 0.00$ | $1.17 \pm 1.15$ | $0.99 \pm 0.00$ |
| Dropout | $1.18 \pm 0.12$ | $\underline{0.92 \pm 0.00}$ | $-1.12 \pm 0.02$ | $0.99 \pm 0.00$ | $3.10 \pm 0.33$ | $0.95 \pm 0.00$ | $-1.20 \pm 0.26$ | $0.98 \pm 0.01$ |
| DUE | $1.96 \pm 0.12$ | $0.79 \pm 0.02$ | $-0.42 \pm 0.31$ | $\mathbf{1.00 \pm 0.00}$ | $1.19 \pm 0.09$ | $0.89 \pm 0.00$ | $-1.49 \pm 0.05$ | $\mathbf{1.00 \pm 0.00}$ |
| EDL | $0.18 \pm 0.03$ | $0.91 \pm 0.01$ | $-1.84 \pm 0.02$ | $\mathbf{1.00 \pm 0.00}$ | $\mathbf{-0.09 \pm 0.03}$ | $0.95 \pm 0.00$ | $-2.08 \pm 0.33$ | $0.99 \pm 0.00$ |
| Ensemble | $1.29 \pm 0.19$ | $\mathbf{0.93 \pm 0.00}$ | $\underline{-2.27 \pm 0.04}$ | $\mathbf{1.00 \pm 0.00}$ | $1.68 \pm 0.24$ | $\mathbf{0.96 \pm 0.00}$ | $\mathbf{-2.52 \pm 0.31}$ | $\mathbf{1.00 \pm 0.00}$ |
| VBLL | $2.64 \pm 1.91$ | $0.89 \pm 0.00$ | $-0.54 \pm 0.24$ | $0.99 \pm 0.00$ | $\underline{-0.04 \pm 0.03}$ | $0.95 \pm 0.00$ | $0.06 \pm 0.89$ | $0.99 \pm 0.00$ |
| LEO (Original) | $\mathbf{0.14 \pm 0.01}$ | $\underline{0.92 \pm 0.00}$ | $\mathbf{-2.58 \pm 0.08}$ | $\mathbf{1.00 \pm 0.00}$ | $-0.06 \pm 0.03$ | $0.95 \pm 0.00$ | $\underline{-2.22 \pm 0.17}$ | $0.99 \pm 0.00$ |
| LEO (KMeans) | $\mathbf{0.12 \pm 0.01}$ | $\underline{0.92 \pm 0.00}$ | $-0.93 \pm 0.40$ | $0.88 \pm 0.07$ | $\underline{-0.04 \pm 0.04}$ | $0.95 \pm 0.00$ | $-1.99 \pm 0.20$ | $0.99 \pm 0.00$ |
| LEO (Random) | $\mathbf{0.12 \pm 0.02}$ | $\mathbf{0.93 \pm 0.00}$ | $-2.55 \pm 0.06$ | $\mathbf{1.00 \pm 0.00}$ | $-0.07 \pm 0.04$ | $0.95 \pm 0.00$ | $-1.37 \pm 1.25$ | $0.99 \pm 0.00$ |

| Dataset | protein | | | | wine | | | |
|---|---|---|---|---|---|---|---|---|
| Metric | NLL ($\downarrow$) | R2 ($\uparrow$) | OOD NLL ($\downarrow$) | OOD R2 ($\uparrow$) | NLL ($\downarrow$) | R2 ($\uparrow$) | OOD NLL ($\downarrow$) | OOD R2 ($\uparrow$) |
| Density R. | $\underline{1.00 \pm 0.23}$ | (*) | $11.64 \pm 2.38$ | $0.39 \pm 0.05$ | $\underline{1.54 \pm 0.06}$ | $-3.83 \pm 5.25$ | $3.27 \pm 0.46$ | (*) |
| Dropout | $4.38 \pm 0.53$ | $\mathbf{0.69 \pm 0.00}$ | $4.82 \pm 0.39$ | $\mathbf{0.54 \pm 0.01}$ | $10.52 \pm 3.03$ | $\mathbf{0.40 \pm 0.02}$ | $12.15 \pm 2.03$ | $\mathbf{0.07 \pm 0.04}$ |
| DUE | $5.12 \pm 0.20$ | $0.08 \pm 0.01$ | $2.99 \pm 0.11$ | $0.14 \pm 0.01$ | $4.35 \pm 0.25$ | $0.11 \pm 0.01$ | $6.10 \pm 0.25$ | $\underline{-0.01 \pm 0.01}$ |
| EDL | $1.07 \pm 0.02$ | $0.41 \pm 0.01$ | $\underline{1.16 \pm 0.07}$ | $0.44 \pm 0.03$ | $2.70 \pm 1.51$ | $\underline{0.36 \pm 0.02}$ | (†) | $-0.04 \pm 0.06$ |
| Ensemble | $2.27 \pm 0.11$ | $0.68 \pm 0.00$ | $1.89 \pm 0.20$ | $0.27 \pm 0.04$ | $5.66 \pm 0.76$ | $0.32 \pm 0.03$ | $4.90 \pm 0.46$ | $-0.28 \pm 0.08$ |
| VBLL | $\underline{1.02 \pm 0.04}$ | $\underline{0.59 \pm 0.01}$ | $2.46 \pm 0.45$ | $-0.20 \pm 0.14$ | $107.26 \pm 100.94$ | $0.29 \pm 0.04$ | $129.77 \pm 78.99$ | $-0.39 \pm 0.10$ |
| LEO (Original) | $\mathbf{0.89 \pm 0.03}$ | $0.60 \pm 0.00$ | $\mathbf{1.16 \pm 0.05}$ | $0.44 \pm 0.02$ | $\mathbf{1.23 \pm 0.03}$ | $\underline{0.37 \pm 0.02}$ | $\mathbf{1.56 \pm 0.03}$ | $-0.03 \pm 0.04$ |
| LEO (KMeans) | $\underline{1.00 \pm 0.05}$ | $0.52 \pm 0.02$ | $\mathbf{1.16 \pm 0.08}$ | $0.42 \pm 0.02$ | $\mathbf{1.23 \pm 0.02}$ | $\underline{0.36 \pm 0.02}$ | $\mathbf{1.54 \pm 0.03}$ | $\underline{0.01 \pm 0.05}$ |
| LEO (Random) | $\mathbf{0.89 \pm 0.03}$ | $0.61 \pm 0.01$ | $\mathbf{1.09 \pm 0.04}$ | $0.47 \pm 0.02$ | $\mathbf{1.23 \pm 0.03}$ | $\underline{0.37 \pm 0.02}$ | $\mathbf{1.56 \pm 0.02}$ | $-0.01 \pm 0.03$ |

# K ILLUSTRATION OF THE EFFECT OF DATA PARTITIONING ON LEO

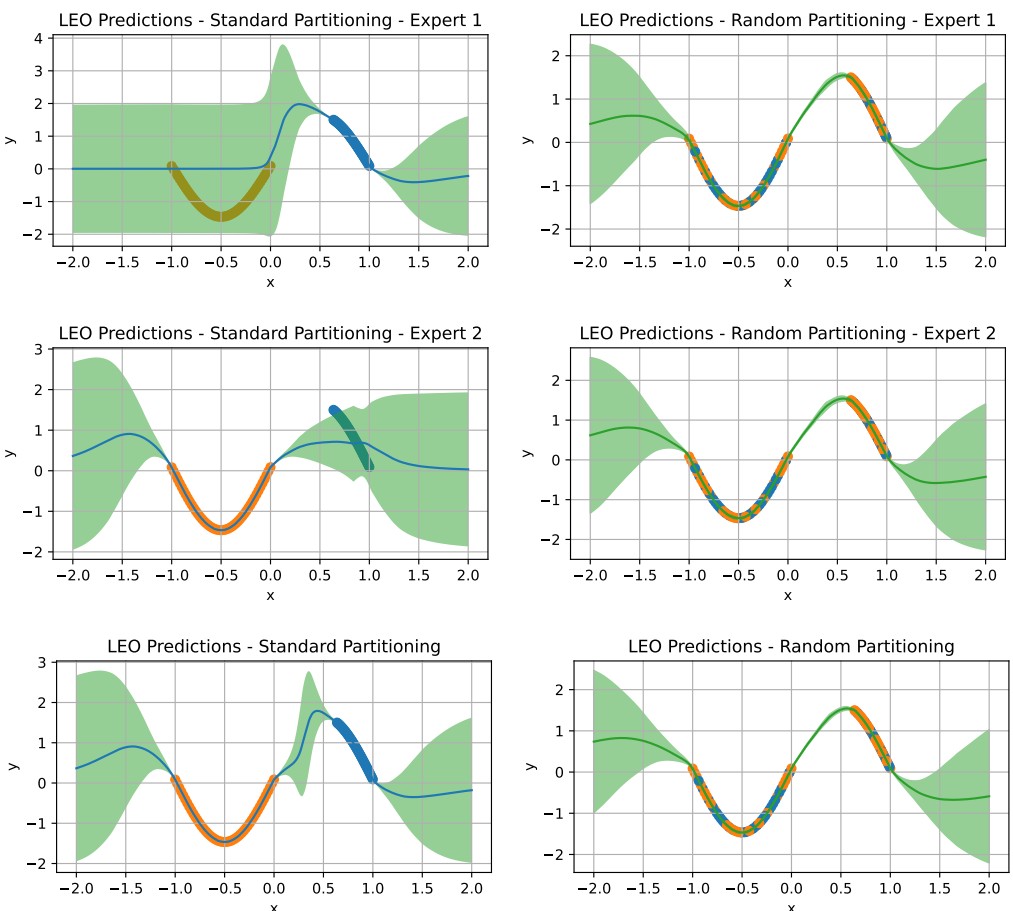

Figure 4: Illustration of the effect of data partitioning on the resulting data fit on a toy problem. Dots indicate training points, and the color indicates which expert they were assigned to. On this toy problem we use two experts and assign datapoints to them using the strategy described in B.1 in the left column and using purely random assignment in the right column. First two rows show the learned model prediction with one expert expert dropped from the model, and the last row shows the resulting full model prediction. As one can easily see, the purely random strategy does not introduce any mismatch between experts' distributions and thus their model fits are nearly identical, which results in poor calibration and is reflected in lack of uncertainty in the middle region $[0, 0.6]$.

## L  ABLATION ON LEO COMPONENTS

Table 12: Ablations results for UCI benchmarks. We compare standard LEO to different versions of LEO with some of its components altered/removed. "Original" refers to standard LEO. "Joint Training" refers to the version where we train experts and router jointly and the gradient from router flows to feature extractor and affects its weights. "no ICV" refers to LEO with router trained without $p(\mathcal{D}_{ICV})$ loss term. "no null expert" refers to version of LEO without the null expert. "RBF distance" refers to version of LEO, where the inverse of L2 distance in the router Equation 2 is replaced with RBF distance (exponent of negative L2 distance). Reported values are means over 20 seeds and the values after $\pm$ denote 95% CIs of the mean estimator. The best methods and all methods that do not statistically differ w.r.t. two-sided z-test are shown in bold. The second best methods are underlined. See Table 6 for explanation of ($\star$) and ($\dagger$) symbols.

| Dataset | boston | | california | | concrete | | energy-efficiency | |
|---|---|---|---|---|---|---|---|---|
| Metric | NLL ($\downarrow$) | R2 ($\uparrow$) | NLL ($\downarrow$) | R2 ($\uparrow$) | NLL ($\downarrow$) | R2 ($\uparrow$) | NLL ($\downarrow$) | R2 ($\uparrow$) |
| Density R. | $0.99 \pm 0.53$ | $0.81 \pm 0.04$ | $0.55 \pm 0.05$ | $0.76 \pm 0.04$ | $0.62 \pm 0.16$ | $0.89 \pm 0.01$ | $1.96 \pm 1.28$ | $0.98 \pm 0.00$ |
| Dropout | $3.75 \pm 0.69$ | $\mathbf{0.87 \pm 0.02}$ | $3.90 \pm 0.25$ | $\mathbf{0.82 \pm 0.01}$ | $2.95 \pm 0.93$ | $\mathbf{0.91 \pm 0.01}$ | $\underline{-0.75 \pm 0.19}$ | $\mathbf{0.99 \pm 0.00}$ |
| DUE | $2.28 \pm 0.38$ | $0.55 \pm 0.10$ | $2.65 \pm 0.13$ | $0.31 \pm 0.07$ | $2.07 \pm 0.15$ | $0.55 \pm 0.09$ | $\underline{-0.80 \pm 0.17}$ | $\mathbf{0.99 \pm 0.00}$ |
| EDL | $\underline{0.46 \pm 0.17}$ | $\mathbf{0.86 \pm 0.02}$ | $\underline{0.49 \pm 0.05}$ | $0.51 \pm 0.53$ | $\underline{0.36 \pm 0.14}$ | $0.90 \pm 0.01$ | $\mathbf{-0.91 \pm 0.09}$ | $0.98 \pm 0.01$ |
| Ensemble | $8.02 \pm 3.06$ | $\mathbf{0.87 \pm 0.02}$ | $2.33 \pm 0.13$ | $0.77 \pm 0.01$ | $4.66 \pm 1.26$ | $\mathbf{0.92 \pm 0.01}$ | $0.34 \pm 0.44$ | $\mathbf{0.99 \pm 0.00}$ |
| VBLL | $3.09 \pm 4.97$ | $\mathbf{0.87 \pm 0.02}$ | $1.31 \pm 0.39$ | $0.71 \pm 0.02$ | $4.31 \pm 3.67$ | $0.90 \pm 0.02$ | $1.75 \pm 1.81$ | $0.98 \pm 0.00$ |
| LEO (Original) | $\mathbf{0.37 \pm 0.16}$ | $0.84 \pm 0.02$ | $0.46 \pm 0.03$ | $\underline{0.79 \pm 0.01}$ | $\mathbf{0.23 \pm 0.08}$ | $\underline{0.89 \pm 0.01}$ | $\mathbf{-0.86 \pm 0.08}$ | $\underline{0.98 \pm 0.00}$ |
| LEO (Joint Training) | $1.84 \pm 0.44$ | $0.81 \pm 0.03$ | $\underline{0.51 \pm 0.03}$ | $0.78 \pm 0.01$ | $0.85 \pm 0.25$ | $0.88 \pm 0.02$ | $-0.53 \pm 0.09$ | $\underline{0.98 \pm 0.00}$ |
| LEO (no ICV) | $\underline{0.45 \pm 0.24}$ | $0.84 \pm 0.02$ | $\mathbf{0.45 \pm 0.02}$ | $\underline{0.79 \pm 0.01}$ | $0.26 \pm 0.11$ | $0.89 \pm 0.01$ | $\mathbf{-0.82 \pm 0.07}$ | $\underline{0.98 \pm 0.00}$ |
| LEO (no null expert) | $1.16 \pm 0.33$ | $0.84 \pm 0.03$ | $9.43 \pm 2.82$ | $0.68 \pm 0.19$ | $0.75 \pm 0.31$ | $\underline{0.89 \pm 0.01}$ | $-0.62 \pm 0.16$ | $0.97 \pm 0.00$ |
| LEO (RBF distance) | $\mathbf{0.34 \pm 0.04}$ | $\underline{0.85 \pm 0.02}$ | $0.56 \pm 0.01$ | $0.78 \pm 0.01$ | $\underline{0.34 \pm 0.02}$ | $0.89 \pm 0.01$ | $-0.05 \pm 0.03$ | $0.97 \pm 0.00$ |

| Dataset | kin8nm | | naval | | power-plant | | yacht | |
|---|---|---|---|---|---|---|---|---|
| Metric | NLL ($\downarrow$) | R2 ($\uparrow$) | NLL ($\downarrow$) | R2 ($\uparrow$) | NLL ($\downarrow$) | R2 ($\uparrow$) | NLL ($\downarrow$) | R2 ($\uparrow$) |
| Density R. | $0.19 \pm 0.03$ | $\underline{0.92 \pm 0.00}$ | $-2.24 \pm 0.05$ | $\mathbf{1.00 \pm 0.00}$ | $\mathbf{-0.09 \pm 0.02}$ | $\underline{0.95 \pm 0.00}$ | $1.17 \pm 1.15$ | $0.99 \pm 0.00$ |
| Dropout | $1.18 \pm 0.12$ | $\underline{0.92 \pm 0.00}$ | $-1.12 \pm 0.02$ | $0.99 \pm 0.00$ | $3.10 \pm 0.33$ | $\underline{0.95 \pm 0.00}$ | $-1.20 \pm 0.26$ | $0.98 \pm 0.01$ |
| DUE | $1.96 \pm 0.12$ | $0.79 \pm 0.02$ | $-0.42 \pm 0.31$ | $\mathbf{1.00 \pm 0.00}$ | $1.19 \pm 0.09$ | $0.89 \pm 0.00$ | $-1.49 \pm 0.05$ | $\mathbf{1.00 \pm 0.00}$ |
| EDL | $0.18 \pm 0.03$ | $0.91 \pm 0.01$ | $-1.84 \pm 0.02$ | $\mathbf{1.00 \pm 0.00}$ | $-0.09 \pm 0.03$ | $\underline{0.95 \pm 0.00}$ | $-2.08 \pm 0.33$ | $0.99 \pm 0.00$ |
| Ensemble | $1.29 \pm 0.19$ | $\mathbf{0.93 \pm 0.00}$ | $\underline{-2.27 \pm 0.04}$ | $\mathbf{1.00 \pm 0.00}$ | $1.68 \pm 0.24$ | $\mathbf{0.96 \pm 0.00}$ | $\mathbf{-2.52 \pm 0.31}$ | $\mathbf{1.00 \pm 0.00}$ |
| VBLL | $2.64 \pm 1.91$ | $0.89 \pm 0.00$ | $-0.54 \pm 0.24$ | $0.99 \pm 0.00$ | $\underline{-0.04 \pm 0.03}$ | $0.95 \pm 0.00$ | $0.06 \pm 0.89$ | $0.99 \pm 0.00$ |
| LEO (Original) | $\mathbf{0.14 \pm 0.01}$ | $\underline{0.92 \pm 0.00}$ | $\mathbf{-2.58 \pm 0.08}$ | $\mathbf{1.00 \pm 0.00}$ | $\mathbf{-0.06 \pm 0.03}$ | $\underline{0.95 \pm 0.00}$ | $\underline{-2.22 \pm 0.17}$ | $0.99 \pm 0.00$ |
| LEO (Joint Training) | $0.31 \pm 0.02$ | $0.90 \pm 0.00$ | $-0.90 \pm 0.05$ | $0.98 \pm 0.00$ | $-0.01 \pm 0.04$ | $0.94 \pm 0.00$ | $-1.66 \pm 0.21$ | $\underline{0.99 \pm 0.00}$ |
| LEO (no ICV) | $\mathbf{0.14 \pm 0.01}$ | $\underline{0.92 \pm 0.00}$ | $\underline{-2.51 \pm 0.11}$ | $\mathbf{1.00 \pm 0.00}$ | $\mathbf{-0.06 \pm 0.03}$ | $\underline{0.95 \pm 0.00}$ | $-1.78 \pm 0.58$ | $\underline{0.99 \pm 0.00}$ |
| LEO (no null expert) | $3.69 \pm 0.48$ | $\underline{0.92 \pm 0.00}$ | $\mathbf{-2.58 \pm 0.09}$ | $\mathbf{1.00 \pm 0.00}$ | $1.16 \pm 0.58$ | $\underline{0.95 \pm 0.00}$ | $-1.83 \pm 0.34$ | $0.99 \pm 0.00$ |
| LEO (RBF distance) | $0.23 \pm 0.01$ | $\underline{0.92 \pm 0.00}$ | $-0.89 \pm 0.48$ | $0.97 \pm 0.03$ | $0.19 \pm 0.02$ | $0.94 \pm 0.00$ | $-0.27 \pm 0.04$ | $0.97 \pm 0.01$ |

| Dataset | protein | | | | wine | | | |
|---|---|---|---|---|---|---|---|---|
| Metric | NLL ($\downarrow$) | R2 ($\uparrow$) | OOD NLL ($\downarrow$) | OOD R2 ($\uparrow$) | NLL ($\downarrow$) | R2 ($\uparrow$) | OOD NLL ($\downarrow$) | OOD R2 ($\uparrow$) |
| Density R. | $1.00 \pm 0.23$ | ($\ast$) | $11.64 \pm 2.38$ | $0.39 \pm 0.05$ | $1.54 \pm 0.06$ | $-3.83 \pm 5.25$ | $3.27 \pm 0.46$ | ($\ast$) |
| Dropout | $4.38 \pm 0.53$ | $\mathbf{0.69 \pm 0.00}$ | $4.82 \pm 0.39$ | $\mathbf{0.54 \pm 0.01}$ | $10.52 \pm 3.03$ | $\mathbf{0.40 \pm 0.02}$ | $12.15 \pm 2.03$ | $\mathbf{0.07 \pm 0.04}$ |
| DUE | $5.12 \pm 0.20$ | $0.08 \pm 0.01$ | $2.99 \pm 0.11$ | $0.14 \pm 0.01$ | $4.35 \pm 0.25$ | $0.11 \pm 0.01$ | $6.10 \pm 0.25$ | $-0.01 \pm 0.01$ |
| EDL | $1.07 \pm 0.02$ | $0.41 \pm 0.01$ | $\mathbf{1.16 \pm 0.07}$ | $\underline{0.44 \pm 0.03}$ | $2.70 \pm 1.51$ | $0.36 \pm 0.02$ | ($\dagger$) | $-0.04 \pm 0.06$ |
| Ensemble | $2.27 \pm 0.11$ | $\underline{0.68 \pm 0.00}$ | $1.89 \pm 0.20$ | $0.27 \pm 0.04$ | $5.66 \pm 0.76$ | $0.32 \pm 0.03$ | $4.90 \pm 0.46$ | $-0.28 \pm 0.08$ |
| VBLL | $1.02 \pm 0.04$ | $\underline{0.59 \pm 0.01}$ | $2.46 \pm 0.45$ | $-0.20 \pm 0.14$ | $107.26 \pm 100.94$ | $0.29 \pm 0.04$ | $129.77 \pm 78.99$ | $-0.39 \pm 0.10$ |
| LEO (Original) | $\mathbf{0.89 \pm 0.03}$ | $0.60 \pm 0.00$ | $\mathbf{1.16 \pm 0.05}$ | $\underline{0.44 \pm 0.02}$ | $1.23 \pm 0.03$ | $0.37 \pm 0.02$ | $1.56 \pm 0.03$ | $-0.03 \pm 0.04$ |
| LEO (Joint Training) | $0.98 \pm 0.05$ | $0.57 \pm 0.01$ | $\underline{1.23 \pm 0.10}$ | $\underline{0.43 \pm 0.03}$ | $1.33 \pm 0.03$ | $0.22 \pm 0.04$ | $1.66 \pm 0.13$ | $-0.03 \pm 0.03$ |
| LEO (ours) (no ICV) | $\mathbf{0.88 \pm 0.03}$ | $0.60 \pm 0.00$ | $1.19 \pm 0.04$ | $\underline{0.43 \pm 0.02}$ | $1.23 \pm 0.03$ | $\underline{0.37 \pm 0.02}$ | $\underline{1.55 \pm 0.02}$ | $-0.02 \pm 0.04$ |
| LEO (no null expert) | $2.57 \pm 0.24$ | $0.58 \pm 0.01$ | $1.69 \pm 0.29$ | $0.31 \pm 0.07$ | $2.38 \pm 0.29$ | $\underline{0.37 \pm 0.02}$ | $2.17 \pm 0.15$ | $-0.36 \pm 0.17$ |
| LEO (RBF distance) | $0.90 \pm 0.01$ | $0.60 \pm 0.00$ | $\underline{1.22 \pm 0.02}$ | $0.35 \pm 0.02$ | $\mathbf{1.16 \pm 0.02}$ | $\mathbf{0.40 \pm 0.01}$ | $\mathbf{1.53 \pm 0.01}$ | $\underline{0.02 \pm 0.02}$ |

## M  COMPARING LEO TO SHALLOW ENSEMBLES

Table 13: Ablations results for UCI benchmarks. We compare standard LEO to equivalent Ensemble models. "LEO (average experts)" refers to version of LEO where the router is completely omitted and the experts are treated as ensemble members and their output is averaged and the variance of their prediction becomes the predictive variance. "Shallow Ensemble" refers to an ensemble, where feature extractor is shared and only heads differ between ensemble members. Reported values are means over 20 seeds and the values after $\pm$ denote 95% CIs of the mean estimator. The best methods and all methods that do not statistically differ w.r.t. two-sided z-test are shown in bold. The second best methods are underlined. See Table 6 for explanation of $(\star)$ and $(\dagger)$ symbols.

| Dataset | boston | | california | | concrete | | energy-efficiency | |
| Metric | NLL ($\downarrow$) | R2 ($\uparrow$) | NLL ($\downarrow$) | R2 ($\uparrow$) | NLL ($\downarrow$) | R2 ($\uparrow$) | NLL ($\downarrow$) | R2 ($\uparrow$) |
|---|---|---|---|---|---|---|---|---|
| Density R. | 0.99 ± 0.53 | 0.81 ± 0.04 | 0.55 ± 0.05 | 0.76 ± 0.04 | 0.62 ± 0.16 | 0.89 ± 0.01 | 1.96 ± 1.28 | 0.98 ± 0.00 |
| Dropout | 3.75 ± 0.69 | **0.87 ± 0.02** | 3.90 ± 0.25 | **0.82 ± 0.01** | 2.95 ± 0.93 | **0.91 ± 0.01** | -0.75 ± 0.19 | **0.99 ± 0.00** |
| DUE | 2.28 ± 0.38 | 0.55 ± 0.10 | 2.65 ± 0.13 | 0.31 ± 0.07 | 2.07 ± 0.15 | 0.55 ± 0.09 | -0.80 ± 0.17 | **0.99 ± 0.00** |
| EDL | **0.46 ± 0.17** | **0.86 ± 0.02** | 0.49 ± 0.05 | 0.51 ± 0.53 | 0.36 ± 0.14 | 0.90 ± 0.01 | **-0.91 ± 0.09** | 0.98 ± 0.01 |
| Ensemble | 8.02 ± 3.06 | **0.87 ± 0.02** | 2.33 ± 0.13 | 0.77 ± 0.01 | 4.66 ± 1.26 | **0.92 ± 0.01** | 0.34 ± 0.44 | **0.99 ± 0.00** |
| VBLL | 3.09 ± 4.97 | **0.87 ± 0.02** | 1.31 ± 0.39 | 0.71 ± 0.02 | 4.31 ± 3.67 | 0.90 ± 0.02 | 1.75 ± 1.81 | 0.98 ± 0.00 |
| LEO | **0.37 ± 0.16** | 0.84 ± 0.02 | **0.46 ± 0.03** | 0.79 ± 0.01 | **0.23 ± 0.08** | 0.89 ± 0.01 | **-0.86 ± 0.08** | 0.98 ± 0.00 |
| LEO (average experts) | 0.79 ± 0.26 | 0.79 ± 0.03 | 1.05 ± 0.05 | 0.41 ± 0.26 | 0.72 ± 0.37 | 0.85 ± 0.02 | -0.23 ± 0.11 | 0.92 ± 0.04 |
| Shallow Ensemble | 678.08 ± 113.27 | **0.87 ± 0.02** | 2379.47 ± 200.84 | 0.77 ± 0.01 | 691.77 ± 254.03 | 0.90 ± 0.01 | 170.36 ± 30.13 | 0.98 ± 0.00 |

| Dataset | kin8nm | | naval | | power-plant | | yacht | |
| Metric | NLL ($\downarrow$) | R2 ($\uparrow$) | NLL ($\downarrow$) | R2 ($\uparrow$) | NLL ($\downarrow$) | R2 ($\uparrow$) | NLL ($\downarrow$) | R2 ($\uparrow$) |
|---|---|---|---|---|---|---|---|---|
| Density R. | 0.19 ± 0.03 | 0.92 ± 0.00 | -2.24 ± 0.05 | **1.00 ± 0.00** | **-0.09 ± 0.02** | 0.95 ± 0.00 | 1.17 ± 1.15 | 0.99 ± 0.00 |
| Dropout | 1.18 ± 0.12 | 0.92 ± 0.00 | -1.12 ± 0.02 | 0.99 ± 0.00 | 3.10 ± 0.33 | 0.95 ± 0.00 | -1.20 ± 0.26 | 0.98 ± 0.01 |
| DUE | 1.96 ± 0.12 | 0.79 ± 0.02 | -0.42 ± 0.31 | **1.00 ± 0.00** | 1.19 ± 0.09 | 0.89 ± 0.00 | -1.49 ± 0.05 | **1.00 ± 0.00** |
| EDL | 0.18 ± 0.03 | 0.91 ± 0.01 | -1.84 ± 0.02 | **1.00 ± 0.00** | **-0.09 ± 0.03** | 0.95 ± 0.00 | -2.08 ± 0.33 | 0.99 ± 0.00 |
| Ensemble | 1.29 ± 0.19 | **0.93 ± 0.00** | -2.27 ± 0.04 | **1.00 ± 0.00** | 1.68 ± 0.24 | **0.96 ± 0.00** | **-2.52 ± 0.31** | **1.00 ± 0.00** |
| VBLL | 2.64 ± 1.91 | 0.89 ± 0.00 | -0.54 ± 0.24 | 0.99 ± 0.00 | -0.04 ± 0.03 | 0.95 ± 0.00 | 0.06 ± 0.89 | 0.99 ± 0.00 |
| LEO | **0.14 ± 0.01** | 0.92 ± 0.00 | **-2.58 ± 0.08** | **1.00 ± 0.00** | -0.06 ± 0.03 | 0.95 ± 0.00 | -2.22 ± 0.17 | 0.99 ± 0.00 |
| LEO (average experts) | 0.92 ± 0.07 | 0.90 ± 0.00 | 0.75 ± 0.23 | 0.26 ± 0.32 | 0.40 ± 0.07 | 0.91 ± 0.01 | -1.17 ± 0.12 | 0.96 ± 0.01 |
| Shallow Ensemble | 341.13 ± 32.74 | 0.90 ± 0.01 | 149.92 ± 19.65 | **1.00 ± 0.00** | 1410.81 ± 174.55 | 0.94 ± 0.00 | 27.52 ± 16.24 | **1.00 ± 0.00** |

| Dataset | protein-w-ood | | | | wine | | | |
| Metric | NLL ($\downarrow$) | R2 ($\uparrow$) | OOD NLL ($\downarrow$) | OOD R2 ($\uparrow$) | NLL ($\downarrow$) | R2 ($\uparrow$) | OOD NLL ($\downarrow$) | OOD R2 ($\uparrow$) |
|---|---|---|---|---|---|---|---|---|
| Density R. | 1.00 ± 0.23 | $(\star)$ | 11.64 ± 2.38 | 0.39 ± 0.05 | 1.54 ± 0.06 | -3.83 ± 5.25 | 3.27 ± 0.46 | $(\star)$ |
| Dropout | 4.38 ± 0.53 | **0.69 ± 0.00** | 4.82 ± 0.39 | **0.54 ± 0.01** | 10.52 ± 3.03 | **0.40 ± 0.02** | 12.15 ± 2.03 | **0.07 ± 0.04** |
| DUE | 5.12 ± 0.20 | 0.08 ± 0.01 | 2.99 ± 0.11 | 0.14 ± 0.01 | 4.35 ± 0.25 | 0.11 ± 0.01 | 6.10 ± 0.25 | -0.01 ± 0.01 |
| EDL | 1.07 ± 0.02 | 0.41 ± 0.01 | **1.16 ± 0.07** | 0.44 ± 0.03 | 2.70 ± 1.51 | 0.36 ± 0.02 | $(\dagger)$ | -0.04 ± 0.06 |
| Ensemble | 2.27 ± 0.11 | 0.68 ± 0.00 | 1.89 ± 0.20 | 0.27 ± 0.04 | 5.66 ± 0.76 | 0.32 ± 0.03 | 4.90 ± 0.46 | -0.28 ± 0.08 |
| VBLL | 1.02 ± 0.04 | 0.59 ± 0.01 | 2.46 ± 0.45 | -0.20 ± 0.14 | 107.26 ± 100.94 | 0.29 ± 0.04 | 129.77 ± 78.99 | -0.39 ± 0.10 |
| LEO | **0.89 ± 0.03** | 0.60 ± 0.00 | **1.16 ± 0.05** | 0.44 ± 0.02 | **1.23 ± 0.03** | 0.37 ± 0.02 | **1.56 ± 0.03** | -0.03 ± 0.04 |
| LEO (average experts) | 1.61 ± 0.10 | 0.28 ± 0.08 | 1.86 ± 0.12 | -1.28 ± 0.91 | 1.73 ± 0.15 | 0.29 ± 0.03 | 1.97 ± 0.10 | -0.62 ± 0.25 |
| Shallow Ensemble | 5732.27 ± 295.16 | 0.56 ± 0.02 | 1269.39 ± 188.26 | 0.46 ± 0.03 | 2836.20 ± 487.40 | 0.33 ± 0.03 | 1421.99 ± 249.10 | -0.39 ± 0.13 |

## N  PROBABILITY OF NULL EXPERTS ASSIGNED BY LEO FOR ID AND OOD EVALUATION SETS

Table 14: Average probabilities of null expert type assigned by LEO model on different in-distribution and out-of-distribution evaluation sets.

| Dataset | Average $p(t \notin \mathcal{E})$ assigned by LEO on eval set |
|---|---|
| protein (ID evalset) | 0.18 |
| protein (OOD evalset) | 0.45 |
| wine (ID evalset) | 0.30 |
| wine (OOD evalset) | 0.60 |
| CIFAR-10 (ID evalset) | 0.0075 |
| CIFAR-10 (OOD evalset) | 0.04 |

## O    LLM USAGE STATEMENT

In preparing this work, we used GPT-5 in three ways: (1) to assist in discovering related literature by suggesting potentially relevant papers, (2) to provide implementation suggestions during development of the experimental code, and (3) to improve clarity of writing. All suggested references were manually checked for correctness and relevance, and all code was reviewed, and verified by the authors.

