# OpenReview forum: "Leave one Expert Out: Robust Uncertainty Quantification via Intrinsic Cross-Validation"
_ICLR.cc/2026/Conference — Submitted to ICLR 2026_

### Official Review · Reviewer_MYNx · 2025-10-31

**Soundness:** 3
**Presentation:** 3
**Contribution:** 2
**Rating:** 2
**Confidence:** 2

**Summary:**

The paper introduces LEO, an architecture and training procedure for enabling models that quantify their epistemic uncertainty. It uses a mixture of shallow experts and a router that assigns examples to experts. The method uses a training procedure that simulates OOD scenarios during training to force the router to fall back on a prior distribution for OOD samples. This design ensures the model remains comparable in size to standard architectures, unlike resource-intensive methods like deep ensembles, which require multiple full models.

**Strengths:**

The paper is interesting and tackles the important problem of epistemic uncertainty quantification in (single) neural networks. The "intrinsic cross-validation" is compelling.

**Weaknesses:**

- The writing is very procedural and lacks motivation and structure. For example, Section 2.2 does not mention that the purpose of the leave-one-expert-out (actually leave multiple experts out!) is to simulate OOD scenarios during training.
- Why randomly corrupted samples in the CIFAR-10 experiments? Why not use e.g. SVHN as Van Amersfoort et al. 2020 does? Did you only evaluate on covariate shift, not on semantic shift?
- Missing citations for claims in the introduction. Examples:
-- Line 34 "assessing the certainty of that prediction remains a notoriously difficult problem" → it all depends on the context. A well calibrated model is very good at estimating the (aleatoric) uncertainty?
-- Line 45: missing citation for aleatoric and epistemic uncertainty
-- Line 67: "The training process typically does not explicitly encourage the model to output high uncertainty in OoD cases." DeepEnsembles try to enforce diversity between the members (data shuffles, different parameter initialization, ...), resulting in dissagreement between the members for OoD inputs.
-- Line 256: No uses of MAE anywhere
-- Table 1 is not referenced in the text
-- Line 299: Are there more recent references to mixture-of-experts literature also related to uncertainty quantification?

**Questions:**

- The router is an OOD detector: it is the sole mechanism that pushes the predicted distribution to the prior. Why is there not an experiment that measures how well p_ϕ(t | x) is able to distinguish OOD from ID? How is the performance of the router affected by the choice of prior? What if the prior does not minimize the loss? In that case, the main mechanism behind LEO explained on line 258 saying that the router collapses to the "vague" prior would not stand?
- Unclear experimental details and lacking insights into and discussion of the results. What do you take the NLL of? Of the probability of Eq 1? How should OOD NLL in Table 2 be interpreted? Is the goal to make the likelihood of the OOD samples high? It is not clear how these numbers show that the model's epistemic uncertainty estimation is good.
- The multiple small "experts" are remeniscent of "ShallowEnsembles" proposed by [1] and benchmarked in [2]. What if you just average the experts' predictions? Is that actually what the router does? How different are the predictions of the experts? The only purpose of the experts is to allow the simulated OOD scenarios during training. Would you agree?

[1]: Lee et al. 2015 https://arxiv.org/abs/1511.06314
[2]: Mucsanyi et al. NeurIPS 2024 https://openreview.net/forum?id=x8RgF2xQTj
[3]: Durasov et al. CVPR'21 https://openaccess.thecvf.com/content/CVPR2021/html/Durasov_Masksembles_for_Uncertainty_Estimation_CVPR_2021_paper.html
[4]: Laurent et al. ICLR'23 https://openreview.net/forum?id=XXTyv1zD9zD

---

> ### Author Response · Authors · 2025-11-18
>
> We would like to thank the reviewer for spending the time to read through our submission and ask insightful questions. We are happy to hear the reviewer finds our paper interesting and the mechanism of ICV compelling.
>
> **Section 2.2 does not mention the purpose of the leave-one-expert-out**
>
> As we wrote at the end of section 2.2:
> “Note that if an expert that is not dropped from the model can extrapolate well to data point types that it did not see during training, the router can achieve a good intrinsic cross-validation likelihood $p_\phi(\mathcal{D}_{\text{ICV}})$ by putting a high probability mass on it, e.g., by setting its temperature $\tau_t$ high. Conversely, if each of the remaining experts makes a wrong prediction, collapsing to the vague prior $p_0$ will be the optimal solution. As such, the router needs to learn its parameters to find the optimal rate at which the model stops trusting the known experts and collapses to the prior, effectively learning how to estimate its epistemic uncertainty. “
>
> We can move this explanation to the beginning of section 2.2 or even earlier in the paper if the reviewer believes this will improve the clarity of writing.
>
> **Why randomly corrupted samples in the CIFAR-10 experiments**
>
> The reviewer is right, we solely considered the problem of covariate shift (differing examples of classes seen during training), as opposed to semantic shift (new classes unseen during training) . The reason for that is we focused on optimising the predictive probability of the developed model and the whole mechanism of intrinsic cross-validation is designed with that objective in mind. As such, we decided to use a corrupted version of CIFAR, as opposed to SVHN, which introduces semantic shifts.
>
> **The router as an OOD detector**
>
> When it comes to covariate shift (as opposed to semantic shift), which is the focus of this paper, there is no clear-cut distinction of OOD vs ID datapoint. If the training domain is between 0 and 1, then should 1.1 be classified as OOD? What about 1.01 or 1.000001? In this case the classification of which datapoint lies out of distribution is more fluid and what matters at the end of the day is how well the model can quantify its uncertainty, which is what we focus on evaluating in the paper. However, per reviewer request we evaluated and reported the average null-expert probabilities assigned by the router on different tasks and displayed them in Appendix N in the revised version of the script. We can see that for every task with an out-of-distribution evaluation set, the average probability assigned to null experts is higher than for the in-distribution evaluation set, which as the reviewer correctly pointed out could be treated as a form of OOD detection, with the aforementioned caveat that in the covariate shift case, labeling points as OOD is slightly more arbitrary than in semantic shift case. However, this signal could still be useful as a rough guidance of how close to training data, the current input point is.
>
> **What if the prior does not minimize the loss?**
>
> The prior does not need to minimise the loss. If none of the experts can make a good prediction then the only way for the model to improve likelihood is by increasing its uncertainty. Collapsing to any one expert will necessarily drive the variance to 0. Conversely, the only way to guarantee high uncertainty is to collapse to the prior, as in regression case each expert is a delta function (zero variance) and null expert (prior) is the only expert with non-zero variance and in classification case, null expert outputs a uniform distribution over classes (maximum possible variance). As such, the collapse to the null expert is tied to increase in uncertainty and must always occur for the calibrated model in situations when none of the experts produces a good fit.
>
> As requested by another reviewer, we ran an ablation, where the null expert has been removed. We display it in Appendix L. We can see that the LEO (no null) variant massively underperforms to LEO on almost every single dataset.
>
> **Unclear experimental details**
>
> Each probabilistic model we use in the paper (LEO and all baselines) returns a distribution over predictions $p(y|x_i)$ for each datapoint $x_i$. As such we simply compute the likelihood of the ground truth under the distribution returned by the model, that is $p(y_i|x_i)$ for a datapoint $(y_i, x_i)$. Using the likelihood of the model as an evaluation metric is standard, and done by every single paper on uncertainty quantification we are aware of.
> OOD NLL simply means computing that negative log-likelihood on the out-of-distribution evaluation set, whereas NLL just refers to negative log-likelihood on the in distribution evaluation set. We will add a short clarification on this to the final version of the paper.

---

> > ### Author Response · Authors · 2025-11-18
> >
> > **Multiple experts and shallow ensembles**
> >
> > As requested by the reviewer, we compared LEO to equivalent shallow ensembles, as well as a version of LEO with no router that just averages the outputs of all experts with equal weights. As such, this version of LEO with no router is same as shallow ensembles, with the exception that each member of shallow ensembles is trained on the exact same dataset, whereas the LEO with no router still uses the data partitioning strategy. We display the results in Appendix M of the revised script. First of all, we see that Shallow Ensembles are the worst performing baseline when it comes to NLL on every single dataset. This is because these shallow models trained on the same dataset tend to be very similar and the variety between them is very small, which causes the variance to collapse almost to zero and effectively eliminate any uncertainty quantification. Compared to Shallow Ensembles, LEO with no router (just averaging experts) performs better, but still massively underperforms to full LEO on almost all datasets, showing that ** the router is not merely averaging all experts ** but instead learning how reliable each expert is and how much to increase its uncertainty as the testpoint gets further away from the training data.
> >
> >
> > **The only purpose of the experts is to allow the simulated OOD scenarios during training.**
> >
> > Yes, generally the reviewer is right that the reason for having multiple experts in our model is so that they can be dropped to simulate an OOD scenario (hence the name of the method and the paper).

---

> > > ### Comment · Reviewer_MYNx · 2025-11-27
> > >
> > > I thank the reviewers for the clarifications and the answers to my questions and weaknesses.  I appreciate the additional experiments and the results.  Therefore, I will raise my score.

---

### Official Review · Reviewer_WJAd · 2025-11-01

**Soundness:** 2
**Presentation:** 2
**Contribution:** 2
**Rating:** 2
**Confidence:** 3

**Summary:**

This paper introduces a method called “Leave on Expert Out” (LEO) which takes a pretrained backbone network $f: \mathcal{X} \to \mathbb{R}^d$ and builds a mixture-of-experts model on-top of it. The paper claims that LEO often produces a mixture-of-experts model that has both good in-distribution prediction performance and relatively good (when compared to other methods) out-of-distribution prediction performance and calibration.

LEO works by training a set of expert prediction heads $h_1, \ldots , h_T: \mathbb{R}^d \to \mathcal{Y}$ on top of $f$, along with a router $r: \mathbb{R}^d \to \Delta^{T + 1}$ that weights the predictions of the experts (with one extra coordinate for a user-defined fallback prediction). Inference on this backbone works in the following way:

1. Given an input $x$, an output prediction $\hat{y}_t = h_t(f(x))$ is generated for each expert $h_t$. These predictions are then converted to output distributions $\delta_1, \ldots, \delta_T$. When doing regression, a scalar prediction $\hat{y}_t$ is converted into a dirac delta function centered at that prediction. When doing classification, predicted logits $\hat{y}_t$ are softmaxed to get a distribution over classes.

2. The final prediction distribution is a mixture of the $T$ expert output distributions along with a single user-specified prior probability distribution $\delta_0$. The weights of the mixture are given by $r(x)$.

   When mixing together dirac delta distributions, in order to preserve differentiability, the output distribution is approximated by a Gaussian whose moments match the true mixture distribution.

LEO training works in three stages:

1. First, each training datapoint is passed through a randomly initialized backbone network to produce a random embedding. These embeddings are then projected onto a random one-dimensional subspace (the same subspace is used for all embeddings), and the datapoints are split up into T contiguous buckets along this subspace, with each contiguous bucket having the same fraction of datapoints assigned to it.

2. T expert heads are created, one for each bucket. Expert-head $t$ is trained to perform well (when attached to the frozen backbone network $f$) at prediction on the $t$th bucket of data.

3. Finally, the router $r$ is trained on two objectives simultaneously: Firstly, the mixture-of-experts model should perform well at in-distribution prediction. Secondly, the mixture-of-experts model should perform well on predicting datapoint $x$’s label, even when the expert corresponding to $x$’s bucket is dropped along with some other random set of experts. When dropping an expert, its output distribution is replaced by the user-specified prior distribution.

   The paper term the mechanism behind this second loss term “intrinsic cross-validation”.

The paper tests LEO in three settings: regression, classification, and performance at black-box bayesian optimization of the Ackley function when LEO is used as a surrogate model to inform UCB what point to sample next. On regression and classification, LEO is comparable to the best performing baseline methods. On optimization of the Ackley function, LEO outperforms baseline methods when optimizing the Ackley function.

**Strengths:**

S1: The LEO algorithm is quite interesting and was novel for me (though admittedly I am somewhat unfamiliar with methods that optimize for good calibration / OOD performance beyond the basics like ensembles). In particular, I liked how the paper was able to utilize a mixture of experts' approach, how they designed their router algorithm which naturally places weight on the prior for really anomalous embeddings, and their intrinsic cross-validation scheme.

S2: I liked that the paper linked to a copy of its codebase at https://anonymous.4open.science/r/leave-one-expert-out-DF01/. Reading through this codebase helped me understand the method and experiments better.

**Weaknesses:**

W1: The evaluations of LEO are all on very toy tasks and none of them involve large scale foundation models. Moreover, the evaluations that are run do not show LEO to be leading other methods by that large of a margin, instead LEO seems just to be among the set of best performing methods.

W2: On the theoretical front, the paper does not really make an attempt to show why LEO should in theory perform better than existing baseline methods. The paper comments on some basic properties of LEO (e.g. lines 166-170), but does not compare its theoretical underpinnings to other methods.

W3: The random-network-embedding-projection type-assignments seem quite arbitrary and not well motivated. I would be curious to compare this approach to some ablations like k-means clustering on trained embeddings, or just completely random assignments.

**Questions:**

Question #1: Regarding weakness W3, what is the motivation behind this approach? Alternatively, could some ablation experiments be run to provide empirical evidence regarding its effect?

Sugggestion #1: The authors could address weakness W1 by running experiments with LEO on large scale foundation models. I would be most interested to see experiments on making foundation models more calibrated at very hard tasks that foundation models only sometimes get right. If LEO beats out baselines here, I would be convinced it is a very good method.

---

> ### Author Response · Authors · 2025-11-18
>
> We would like to thank the reviewer for spending the time to read our submission and suggest interesting new experiments. We are glad to hear the reviewer finds our method novel and interesting. We address all questions/concerns below.
>
> **Weakness 1**
>
> We would not agree that our experiments are all toy tasks, as all of our experiments are real-world benchmarks. These benchmarks are standard and were used in preceding literature when evaluating OOD behaviour of models. We agree that applying LEO to foundational models constitutes an interesting direction of future work, however, we would not say that our experiments are only small-scale. Datasets such a CIFAR (50K of 32x32 images) or UTK-Face (20K of 200x200 images) are mid-to-large scale and when conducting experiments on them, we used the ResNet and WideResNet models. We also believe that compare to most recent papers in the field of uncertainty quantification in deep learning, our results are on a comparable scale. For example, in VBLL (ICLR 2024) the largest end-to-end trained experiment is CIFAR. In DensityRegression (AISTATS 2024) the largest end-to-end trained experiment is a computer vision depth-estimation problem, comparable in scale to our UTK-Face experiment. As such, we believe the scale and scope of our experiments, generally matches the one of similar papers.
>
> When it comes to performance of LEO, it is true that the margin of how much LEO outperforms existing methods is not always very large, but as reviewer correctly pointed out, LEO is consistently among top performing methods. However, we believe that this consistency is the strongest point of LEO. Indeed, on tasks where optimal predictive uncertainty is large, an overconfident model will underperform, but conservative models that output high uncertainty no matter what will excel. The opposite is true as well. As such, the fact that LEO can perform well across all benchmarks means it is able to correctly calibrate its predictive uncertainty.
>
>
> **Weakness 2**
>
> As we wrote at the end of section 2.2:
> “Note that if an expert that is not dropped from the model can extrapolate well to data point types that it did not see during training, the router can achieve a good intrinsic cross-validation likelihood $p_\phi(\mathcal{D}_{\text{ICV}})$ by putting a high probability mass on it, e.g., by setting its temperature $\tau_t$ high. Conversely, if each of the remaining experts makes a wrong prediction, collapsing to the vague prior $p_0$ will be the optimal solution. As such, the router needs to learn its parameters to find the optimal rate at which the model stops trusting the known experts and collapses to the prior, effectively learning how to estimate its epistemic uncertainty. “
>
>
> As such, our method, in contrast to every other, explicitly simulates an OOD scenario and trains the model to perform well in that simulated scenario. Other methods either introduce some inductive biases (DUE, DUQ, Density Regression) to increase uncertainty far from training data, perturb the model in some way (Dropout, Ensemble) or rely on some regulariser (VBLL, EDL) to ensure model uncertainty does not collapse. However, none of the methods explicitly trains the model of epistemic uncertainty to perform well in OOD scenarios. By dropping an expert and simulating such a scenario, our method explicitly optimised for the metric of interest, which is reflected in robust performance.
>
>
> **Weakness 3**
>
> As requested by the reviewer we ran additional experiments and compared different strategies of assigning the types to datapoints, which are available in the revised version of the submission, in Appendix J.  We compared against performing k-means clustering and purely random assignment. We can see that although on many datasets the performance is similar, there are clear failure cases for alternative strategies (e.g. yacht or naval datasets).
>
> In Appendix K, we also present a new Figure that explains why mismatch between distributions is important on a 1D toy problem and how the purely random assignment can easily fail.

---

### Official Review · Reviewer_r2SE · 2025-11-02

**Soundness:** 3
**Presentation:** 4
**Contribution:** 4
**Rating:** 6
**Confidence:** 4

**Summary:**

This paper proposes “Leave one Expert Out” (LEO) for uncertainty quantification in deep learning. LEO has a mixture of experts architecture, where the router uses the latent space representation of input to understand its distance from the training data to weight the expert predictions. It employs a null expert to collapse the model output to a prior if the input is different from the training data. The model is trained using a new mechanism called intrinsic cross-validation (ICV). ICV drops one or more experts during training, simulating out-of-distribution cases, which results in epistemic uncertainty quantification.

**Strengths:**

- The paper is well written and organized.
- The proposed method is novel.
- The proposed method is well motivated with connections to well established theory from Gaussian Processes and Bayesian methods.
- Uncertainty quantification is an important field for safety critical applications of deep learning.
- The empirical results are compelling, providing advancements on state-of-the-art in uncertainty quantification.

**Weaknesses:**

- The empirical results are limited:
   - Ablations on main components like ICV, null expert, different distance metrics are needed to understand the contributions of each.
   - The router has hyperparameters and it would be good to showcase the sensitivity of results to these parameters. Similarly, sensitivity on the number of experts is missing.
   - The initial “type” assessment mentioned in the appendix B1 seems random and might be subject to variability. It would be good to provide experiments showing how choices here affect the results.
- The experiments are conducted at small scales and for certain tasks. Applying the method to text domains and LLMs would strengthen the paper.

**Questions:**

I mention my main questions and concerns in the weakness section. Some minor points:
- Figure 2b. Most of the methods seems to coincide and it is hard to understand the individual performance of the methods. Is LEO’s CI’s for OOD include ensemble's CI for OOD? If so, I’d love to hear author’s interpretation of the superior performance of ensemble method for both ID and OOD.
- Table 4:
   - I’d recommend adding the deterministic model’s performance numbers for comparison.
   - Is it possible to provide a similar table for training requirements?

---

> ### Author Response · Authors · 2025-11-18
>
> We would like to thank the reviewer for spending time to read our submission and for suggesting interesting ablations. We are happy to hear the reviewer finds our method novel and well-motivated. We address the reviewer’s questions and concerns below.
>
> **Sensitivity to hyperparameters**
>
> We would like to emphasise that the parameters of the router (temperature, calibration constants) **are not hyperparameters**. These parameters are learnt by the router during the router training loss, while utilising the intrinsic cross-validation mechanism. As such, they are selected automatically and do not need to be specified by the user.
>
>
> **Ablation on number of experts**
>
> As requested by the reviewer, we ran an additional ablation on the number of experts on the UCI benchmarks and displayed them in Appendix I in revised script. We compare models with 2, 3, 5 and 10 experts. Of course having extremely small number of experts (like 2) can result in the OOD signal being too weak, whereas having too many (like 10) can result in each experts being able to see very few training points especially on datasets like yacht or energy-efficient, that have very small training set sizes.
>
> However, we want to emphasise that on many datasets, LEO achieves the best performance with all tested number of experts and even in the cases where extremely small or large number of experts cause underperformance, it still places that variant of LEO among second-best performing methods, which signifies good degree of robustness to this choice. We also believe that the number of experts we selected (5) and used across all of the experiments, is a universal choice that works well across different domains.
>
>
> **Ablations on different components**
>
> As requested by the reviewer, we ran additional ablations on different components of LEO and display it in Appendix L in the revised version of the script. The versions of LEO we compare against are: no ICV, no null expert and RBF distance (exponent of negative squared distance) instead of the inverse of L2 distance we use.
> We see that except for the naval dataset, removing the null expert from the model drastically harms the performance. Switching to RBF distance, which collapses much faster to zero, due to the presence of exponent, also harms the performance on the majority of datasets, nonetheless retains relatively good performance in the OOD cases. This makes sense as in OOD case, we generally want to collapse to the prior fast, however, this is suboptimal when evaluation samples are in-distributions. As such, we believe our choice of the inverse L2 distance strikes a good balance and allows the router to calibrate itself in a way that maintains good performance in both OOD and ID cases. Removing the ICV does not result in significant performance drop if a large validation set is available (since for all methods and experiments we use a validation set for early stopping as described in experimental details), so the training of the router can be stopped at the moment where the pace of collapse to the prior is optimal. However, on small datasets such as boston and yacht, we cannot afford to set aside a large validation set and thus instead of standard cross-validation, relying on intrinsic cross-validation is preferable. As such, we see that on these datasets, removing the ICV mechanism results in a significant performance drop.
>
>
> **Ablation on type assignment**
>
> As requested by the reviewer we ran additional experiments and compared different strategies of assigning the types to datapoints, which are available in the revised version of the submission, in Appendix J.  We compared against performing k-means clustering and purely random assignment. We can see that although on many datasets the performance is similar, there are clear failure cases for alternative strategies (e.g. yacht of naval datasets).
>
> In Appendix K, we also present a new Figure that explains why mismatch between distributions is important on a 1D toy problem and how the purely random assignment can easily fail.

---

> ### Author Response · Authors · 2025-11-18
>
> **Good performance of Ensembles on CIFAR10**
>
> The reviewer is right that when it comes to ECE scores, displayed in Figure 2b), CIs of LEO and Ensembles overlap in the OOD case, but this is no longer the case when measuring their performance according to NLL loss. Expected Calibration Error (ECE) does not measure whether a model’s predictions are correct, but rather whether its predicted confidence matches the frequency of being correct. Because ECE uses histogram binning, it is a relatively coarse metric: extreme overconfident errors are averaged within a bin and therefore have limited influence on the final score.
> In contrast, the full predictive log-likelihood is highly sensitive to overconfident mistakes. When comparing models via likelihood, the confidence intervals no longer overlap, and LEO clearly outperforms Ensembles. This indicates that although both models may appear similarly calibrated under ECE, the Ensembles model is in fact more overconfident, and its overconfidence is revealed by the stronger, more sensitive likelihood metric.
>
> **Running times in Table 4**
>
> In the updated version of the script, we expanded Table 4 and provided the running times and memory requirement for a baseline model and also included the training time for each method. For training time, we included the total time needed for validation loss to converge, while using the patience mechanism we described in the training details (for CIFAR, we finish training if the validation loss fails to increase for 10 epochs). We see that the base model has, of course, the shortest training time, followed by LEO, which is the fastest method among all uncertainty quantification methods, highlighting another benefit of our method over the competitors.
>
> **Scale of experiments**
>
> We agree that applying LEO to large-scale models such as LLMs constitutes an interesting direction of future work, however, we would not say that our experiments are only small-scale. Datasets such a CIFAR (50K of 32x32 images) or UTK-Face (20K of 200x200 images) are mid-to-large scale and when conducting experiments on them, we used the ResNet and WideResNet models. We also believe that compare to most recent papers in the field of uncertainty quantification in deep learning, our results are on a comparable scale. For example, in VBLL (ICLR 2024) the largest end-to-end trained experiment is CIFAR. In DensityRegression (AISTATS 2024) the largest end-to-end trained experiment is a computer vision depth-estimation problem, comparable in scale to our UTK-Face experiment. As such, we believe the scale and scope of our experiments, generally matches the one of similar papers.

---

### Official Review · Reviewer_orKS · 2025-11-05

**Soundness:** 2
**Presentation:** 2
**Contribution:** 3
**Rating:** 2
**Confidence:** 4

**Summary:**

This paper proposes a single model approach to epistemic uncertainty estimation. The training samples are partitioned into disjoint sets, and during training each set is assigned to a specific expert (linear head); all experts share the same feature extractor. Some key steps are the inclusion of a null/prior expert, the training of a router module which learns the effect of expert exposure to unknown samples, and the overall two-phase training procedure denoted as intrinsic cross-validation (ICV). The system thus learns a supervised OoD signal from the training data alone, while remaining within a single-model framework.

**Strengths:**

- the ICV mechanism tweaks elegantly the mixture-of-experts paradigm in order to create OoD scenarios within the training set in a supervised manner (i.e., by increasing the probability of the null expert); this contrasts with methods like DUQ which rely on latent distance in an implicit manner

- low memory footprint and good inference speed

- the method is validated on three distinct UQ domains, with a very good performance on the BO experiment which matches/outperforms in high dimensions the GP

**Weaknesses:**

- the data partitioning strategy is badly justified, and seems arbitrary and plainly wrong with respect to the intended assumption : that the experts are specialists on *distinct* data partitions. The strategy passes samples across a freshly initialized, untrained data extractor and projects embeddings on a random vector. There is no semantic meaning whatsoever, this seems to me like a random vector projection of a random space projection, highly unlikely to create semantically coherent factors. The paper claims this creates a mismatch across experts' training distributions, but it is an arbitrary mismatch. The fact the method works well in these circumstances raises a critical question, namely is the method's strong performance a result of this very high variance partitioning setup, and if yes, why? Since it goes in my opinion against the distinct data partition assumption. Overall, this part is critical and is awfully under studied for the moment.

- an ablation focusing on the number of experts is necessary; the considered number (five) which matches the number of ensemble models / droupout samples is reasonable for comparison purposes, but it is in no way a methodologically sound justification. The study is crucial since the two pitfalls I can foresee are significant : too few experts and the OoD signal becomes weak, too many experts and the method might exhibit underfitting and poor generalization.

- while convenient, the two step training introduces a limitation; the latent space is good for the expert loss (CE) but it is never trained to produce a latent space useful for the router's task which uses a L2 metric in the aforementioned space.  Why dismiss  end-to-end training completely? The justification that it is this "much faster" is of convenience, and should be underlined rather as a limitation, and justified better following a comparison with end-to-end training

- the claims of performance against SOTA are overstated and should be toned down (e.g. Figure2 Ensemble on ID NLL, ECE and DUQ/EDL on OoD NLL). LEO is good at both which places it conveniently on the Pareto front, but it is a trade-off, it does not blow away the competition.

**Questions:**

- justify formally the link between the data partitioning strategy and the claimed emergence of distinct data partitions

- a characterization of the choice of the number of experts which would sustain the emergence of the OoD signal

- a justification for the choice of discarding the option of end-to-end training

**Details Of Ethics Concerns:**

nothing in particular

---

> ### Author Response · Authors · 2025-11-18
>
> We would like to thank the reviewer for spending the time to read our paper and for asking insightful questions. We are happy to hear the reviewer finds our ICV mechanism elegant and appreciates the performance of our method on the BO domain. We address questions and concerts below.
>
>
> **Data Partitioning**
>
> The reviewer is right that the mismatch in expert’s training partitions is an arbitrary mismatch. However, our method does not require semantically meaningful mismatch between distributions. Instead, we require it to have some form of mismatch, so that the router can judge how far each of the experts can reliably extrapolate and learn how to translate the distance in latent space between current example and expert’s centroid to uncertainty in the output space. While relying on some form of semantic similarity might also be an interesting solution, requiring access to such semantic similarity metric would limit the applicability of the method to arbitrary tasks. As such, we opted for such a task-agnostic partitioning method.
>
> Studying formally the shape of emerging partitions is challenging, but nonetheless possible if we were to assume the network is large enough so that its behaviour is close enough to the infinitely large neural network, which has been studied by [1]. That work explicitly derived the covariance of two neurons $z^{(l)}$, $z^{\prime (l)}$ for two different inputs $x$, $x^\prime$ to the network at layer $l$ denoted as $K^{(l)}(x,x^\prime)$. For example, in case of ReLU activations, the equation for correlation is given in their equation 11 and it is essentially a very complex function of cosine similarities of the inputs in the original input space. When we then project the latent embeddings of $n$ such neurons with a random vector $v \in \mathbb{R}^{n}$ such that $v \sim \mathcal{N}(0, \mathbb{I})$, as we do in our work, we get that the resulting covariance is:
>
> $$
> \textrm{Cov}(v^T z^{(l)}, v^T z^{\prime(l)}) = \sum_{i=1}^n \textrm{Cov}(v_i z^{(l)}_i, v_i z^{\prime(l)}_i) = n  K^{(l)}(x,x^\prime)
> $$
>
> where the first inequality is true as different neurons will be independent (see [1], end of page 3). As such, the correlations after projections are maintained. As such, our partitioning strategy, will generally create partitions by similarity measured by $K^{(l)}(x,x^\prime)$, which is a complicated function of cosine similarities of $x$ and $x^{\prime}$. While this does not guarantee a semantic mismatch, this will generally mean inputs that are similar in input space will also be on average similar in the (freshly initialised) latent space.
>
> To give more insights on how different partitioning strategies would influence the model performance, we ran ablations on the UCI datasets, which are available in the revised version of the submission, in Appendix J.  We can see that although on many datasets the performance is similar, there are clear failure cases for alternative strategies (e.g. yacht of naval datasets).
> In Appendix K, we also present a new Figure that explains why mismatch between distributions is important on a 1D toy problem and how the purely random assignment can easily fail.
>
>
> [1] Lee, Jaehoon, et al. "Deep neural networks as gaussian processes." arXiv preprint arXiv:1711.00165 (2017).
>
> **Ablation on number of experts**
>
> As requested by the reviewer, we ran an additional ablation on the number of experts on the UCI benchmarks and displayed them in Appendix I in revised script. We compare models with 2, 3, 5 and 10 experts. The reviewer is right that having too few experts (such as 2), can cause the OOD signal to become too weak and the model to underperform, which we observe on e.g. the power-plant dataset. At the other side of the spectrum if the number of experts is too big, especially on small datasets, each expert will only see a very small portion of training data causing underperformance, which we observe on smaller datasets such as yacht and energy-efficient.
>
> However, we want to emphasise that on many datasets, LEO achieves the best performance with all tested number of experts and even in the cases where extremely small or large number of experts cause underperformance, it still places that variant of LEO among second-best performing methods, which signifies good degree of robustness to this choice. We also believe that the number of experts we selected (5) and used across all of the experiments, is a universal choice that works well across different domains.

---

> ### Author Response · Authors · 2025-11-18
>
> **Comparing to end-to-end training**
>
> During the router training, the router calibrates itself to translate the distance in latent space to the uncertainty over output. This is done over the latent space produced by expert training, over which the router has no control. We argue that this is actually beneficial, as in this way the OOD signal is completely separated from the expert training. Otherwise, the OOD loss used to train the router influences how the latent space is formed, which in turn influences how the experts learn. We believe this could create some form of information leakage, and the simulated OOD scenarios can stop being actually OOD and the whole underpinning idea can break.
>
> We ran an additional experiment confirming that. We display it in Appendix L. The name “LEO (joint training, full gradient)” refers to a jointly trained LEO, where the gradient from the router is propagated to feature extractor. We see that it underperforms to standard LEO on all datasets.
>
>
> **Claims against SOTA**
>
> We agree that stating “[LEO] completely outperforms existing methods ” in introduction, might be too strong and we decided to change it to "consistently matches or outperforms state of art performance”.

---

### Author Response · Authors · 2025-11-18

We would like to thank all the reviewers for spending time to read through our submission and provide useful feedback. We have now updated the submission and added Appendices I - N at the end of the document. These include the requested ablation, as well as a new Figure explaining the intuition behind the proposed data partitioning strategy. We would be most grateful to hear back from the reviewers before the end of discussion period.

---

### Meta-Review · Area_Chair_frBu · 2025-12-26

**Summary:**

The paper proposes Leave One Expert Out (LEO), a mixture-of-experts framework for uncertainty quantification. The core novelty is a training mechanism, Intrinsic Cross-Validation (ICV), where experts are temporarily dropped during training to simulate Out-of-Distribution (OOD) scenarios. A router learns to assign weights to the remaining experts or collapse to a "null expert" (representing a prior belief) when the input is sufficiently distinct from the training partitions. The authors evaluate LEO on UCI, CIFAR-10, and black-box optimization tasks.

Reviewers generally agree that the LEO algorithm is a novel conceptual contribution. Primary concerns are the arbitrary nature of the data partitioning (which undermines the expert specialization premise), the use of small-scale/toy benchmarks such as UCI, CIFAR-10, and the overstatement of performance against SOTA.

Overall, I think the paper is borderline, and future iterations should expand both the scale and scope of experiments to address these concerns.

**Reviewer Concerns:**

**Addressed concerns**
- The rebuttal provided requested ablation experiments in Appendices regarding the number of experts and the necessity of the null expert, and expanded runtime/training cost reporting.
- An additional experiment was provided to demonstrate that end-to-end training degrades performance, justifying the two-stage pipeline choice.
- Several points of clarifications were provided in response to reviewer questions.

**Outstanding concerns**
- A major concern is that the expert partitioning strategy appears largely arbitrary rather than meaningful specialists as they are assigned through projecting embeddings on a random vector. The authors' rebuttal, relying on infinite-width network theory to claim these projections maintain input correlations, offers interesting insights but still fails to explain the experts are true specialists. They are simply training on random slices of data grouped by an arbitrary similarity function. The concern that the method's performance might be due to high-variance randomness rather than the proposed expert logic. New ablations in Appendix J further confirms that the performances of LEO are generally strong regardless of the partitioning strategies.
- Multiple reviewers criticized the evaluation on small-scale/toy datasets. While the authors argued that these are standard for the UQ literature, I believe this prevents the paper from demonstrating that the approach meaningfully captures more challenging or realistic forms of epistemic uncertainty, and the true impact of this work remains unclear.
- LEO is often just among the set of best performing methods rather than a clear winner. While the authors toned down one sentence in the introduction, I think a more comprehensive revision of claims regarding performance against SOTA should be performed.

**Reviewer Scores:**

- Reviewer orKS is likely to maintain the score. The rebuttal does not address the concern that the "Mixture of Experts" is misleading when the experts are arbitrarily assigned.
- Reviewer r2SE is likely to maintain the initial positive assessment.
- Reviewer WJAd is likely to maintain the score because the authors did not run the requested large-scale experiments.
- Reviewer MYNx is likely to increase the score to borderline as most of the concerns were about clarity and the authors adequately addressed them.

---

### Decision · Program_Chairs · 2026-01-26

Reject